# Unfolded protein-independent IRE1 activation contributes to multifaceted developmental processes in Arabidopsis

Kei-ichiro Mishiba ⓘ, Yuji Iwata, Tomofumi Mochizuki, Atsushi Matsumura, Nanami Nishioka, Rikako Hirata, Nozomu Koizumi

**In Arabidopsis, the *IRE1A* and *IRE1B* double mutant (*ire1a/b*) is unable to activate cytoplasmic splicing of *bZIP60* mRNA and regulated IRE1-dependent decay under ER stress, whereas the mutant does not exhibit severe developmental defects under normal conditions. In this study, we focused on the Arabidopsis *IRE1C* gene, whose product lacks a sensor domain. We found that the *ire1a/b/c* triple mutant is lethal, and heterozygous *IRE1C* (*ire1c/+*) mutation in the *ire1a/b* mutants resulted in growth defects and reduction of the number of pollen grains. Genetic analysis revealed that *IRE1C* is required for male gametophyte development in the *ire1a/b* mutant background. Expression of a mutant form of IRE1B that lacks the luminal sensor domain (ΔLD) complemented a developmental defect in the male gametophyte in *ire1a/b/c* haplotype. In vivo, the ΔLD protein was activated by glycerol treatment that increases the composition of saturated lipid and was able to activate regulated IRE1-dependent decay but not *bZIP60* splicing. These observations suggest that IRE1 contributes to plant development, especially male gametogenesis, using an alternative activation mechanism that bypasses the unfolded protein-sensing luminal domain.**

## Introduction

The ER in eukaryotes copes with an accumulation of unfolded proteins by activating the unfolded protein response (UPR), which increases protein folding capacity and attenuates protein synthesis in the ER (Walter & Ron, 2011). Inositol-requiring enzyme 1 (IRE1) is the primary transducer of the UPR. IRE1 consists of an N-terminal sensor domain facing the ER lumen, a single transmembrane helix embedded in the ER membrane, and kinase and RNase domains at its C terminus on the cytosolic side (Nikawa & Yamashita, 1992; Sidrauski & Walter, 1997). Under ER stress, IRE1 senses ER luminal unfolded proteins, ultimately leading to IRE1 dimerization, autophosphorylation,

and RNase activation, which catalyze cytoplasmic splicing. Targets of the cytoplasmic splicing are mRNAs encoding UPR-specific transcription factors, such as HAC1 in yeasts (Sidrauski & Walter, 1997), XBP1 in metazoans (Yoshida et al, 2001), and bZIP60 in Arabidopsis (Deng et al, 2011; Nagashima et al, 2011). Activated IRE1 also degrades mRNAs encoding secretory pathway proteins, designated as the regulated IRE1-dependent decay (RIDD) of mRNAs in fission yeast (Kimmig et al, 2012), metazoans (Hollien & Weissman, 2006; Iqbal et al, 2008; Han et al, 2009; Hollien et al, 2009), and plants (Mishiba et al, 2013; Hayashi et al, 2016). Although distinct catalytic mechanisms between cytoplasmic splicing and RIDD has been reported (Tam et al, 2014), how IRE1 outputs these two modules during physiological and developmental processes is still unclear (Maurel et al, 2014).

Although IRE1-deficient mice (Zhang et al, 2005) and flies (Ryoo et al, 2013) cause embryonic lethality, IRE1-deficient yeast (Nikawa & Yamashita, 1992; Kimmig et al, 2012) and worms (Shen et al, 2001) are viable. In plants, Arabidopsis IRE1A- and IRE1B-defective mutants do not exhibit severe developmental phenotypes under normal conditions (Nagashima et al, 2011; Chen & Brandizzi, 2012), whereas rice homozygotes that express kinase-defective IRE1 is lethal (Wakasa et al, 2012; note that rice has one *IRE1* gene). The disparate phenotypic consequences of IRE1 mutation between Arabidopsis and rice prompted us to investigate the degree of contribution that IRE1 makes to plant development.

In recent years, activations of IRE1 caused by lipid perturbation or inositol depletion were observed in yeast (Pineau et al, 2009; Promlek et al, 2011; Lajoie et al, 2012), human cells (Ariyama et al, 2010), and mouse cells (Volmer et al, 2013). These IRE1 activations do not require sensing of unfolded proteins by the luminal domain of IRE1 (Snapp, 2012) but does require an amphipathic helix adjacent to the transmembrane helix to sense ER membrane aberrancies (Halbleib et al, 2017). Although physiological functions of the alternative IRE1 activation are less well known, it has been presumed that unfolded protein-independent mechanisms allow cells to preemptively adapt their ER folding capacity (Volmer & Ron, 2015).

---

Graduate School of Life and Environmental Sciences, Osaka Prefecture University, Osaka, Japan

Correspondence: mishiba@plant.osakafu-u.ac.jp

For instance, mutant worms with decreased membrane phospholipid desaturation activate IRE1 without promoting unfolded protein aggregates (Hou et al, 2014). However, there are no studies directly addressing the importance of the unfolded protein-independent IRE1 activation in developmental processes in multicellular organisms.

In this report, we investigated the contribution of IRE1 lacking its sensor domain to Arabidopsis development. We found that a third Arabidopsis *IRE1* gene, encoding sensor domain-lacking IRE1, is functional and that the triple mutant of the three *IRE1* (*IRE1A-C*) genes is lethal. Our analyses with plants that express mutant IRE1B proteins without the sensor domain suggest contribution of unfolded protein-independent IRE1 activation to multifaceted developmental processes in Arabidopsis.

# Results

## Loss of function of IRE1C does not alter ER stress response

In addition to the *IRE1A* and *IRE1B* genes, Arabidopsis contains an *IRE1-like* gene (AT3G11870; designated as *IRE1C* hereafter), whose product lacks a sensor domain (Fig 1A). The sensor domain–lacking IRE1 was also found in some other Brassicaceae species, such as *Camelina sativa*, and phylogenetic analysis showed that the IRE1C forms an independent cluster from IRE1A and IRE1B groups in dicotyledonous plants (Fig 1B). A T-DNA insertion mutant of *ire1c* (SALK_204405; Fig S1A) and *ire1a ire1c* (designated as *ire1a/c* hereafter) double mutants did not exhibit any visible phenotypic alterations in normal growth conditions (Fig 1C). Consistent with the previous studies (Nagashima et al, 2011; Mishiba et al, 2013), susceptibilities to ER stress inducers, DTT, and tunicamycin (Tm) were more apparent in *ire1a/b* mutant than those in WT, *ire1a*, and *ire1b* mutants (Figs 1D and S1B and C). The susceptibilities to DTT and Tm in *ire1c* and *ire1a/c* mutants were same as that in WT (Figs 1D and S1B and C). To detect *bZIP60* splicing and RIDD in the *IRE1* mutants under ER stress, expressions of *BiP3* and *PR-4* mRNA, which are the typical targets of bZIP60 (Iwata & Koizumi, 2005) and RIDD (Mishiba et al, 2013), respectively, and the spliced form of *bZIP60* (*bZIP60s*) mRNA were analyzed by qPCR. Up-regulation of *BiP3* and *bZIP60s* mRNA and down-regulation of *PR-4* mRNA by Tm and DTT treatments were observed in *ire1c* and *ire1a/c* mutants as well as WT, *ire1a*, and *ire1b* mutants, but not in *ire1a/b* mutant (Fig 1E). These results indicate that IRE1C, which lacks a sensor domain, does not contribute to ER stress response in Arabidopsis.

## A triple mutant of *IRE1A*, *IRE1B*, and *IRE1C* is lethal

We tried to produce *ire1a/b/c* triple mutant by crossing *ire1a/b* with *ire1c* mutants. We obtained three F$_2$ plants heterozygous for *ire1c* (designated as *ire1c/+*) and homozygous for *ire1a* and *ire1b*. Genotyping of their self-pollinated progenies showed that no plants homozygous for *ire1c* were obtained among 108 plants analyzed (Table 1 and Fig S1D). In addition, unexpected segregation of homozygotes and heterozygotes for *IRE1C* (+/+:*ire1c/+* = 1.0:0.3) was

observed. These results indicate that *ire1a/b/c* triple mutant is lethal. The *ire1a/b ire1c/+* plants exhibited growth retardation (Fig 2A) and reduced seed set (Fig 2B and C) compared with the *ire1a/b* +/+ siblings. Pollen development was especially impaired in the *ire1a/b ire1c/+* plants, whereas *ire1c* and *ire1a/c* mutants did not affect pollen development and seed set (Fig 2C). No transmission of the *ire1c* allele through male gametophyte was shown after reciprocal crossing between *ire1a/b* and *ire1a/b ire1c/+* mutants (Table 1). Transgenic plants carrying *IRE1C* promoter-driven *GUS* (β-glucuronidase) reporter gene construct (Fig S2A) showed that IRE1C is expressed in the anther (Fig S3A) and embryo (Fig S3B). No visible GUS staining was observed in vegetative tissues (root, leaf, and stem) of young seedlings with or without stress treatments (Fig S3C). These observations are qualitatively consistent with the microarray database (Arabidopsis eFP browser; http://bar.utoronto.ca/efp/cgi-bin/efpWeb.cgi), which shows that the *IRE1C* gene is scarcely expressed in vegetative tissues. Growth retardation of *ire1a/b ire1c/+* plants was also observed under in vitro culture containing 1% sucrose (Fig S3D), under which *IRE1C* mRNA was slightly decreased in *ire1a/b ire1c/+* compared with *ire1a/b* (Fig S3E). The decrease in *IRE1C* mRNA level in *ire1a/b ire1c/+* was prominent in flower bud tissues (Fig S3F). Taken together, IRE1C, which lacks a sensor domain, contributes to male gametophyte development and acts redundantly with IRE1A and IRE1B.

## Mutant IRE1B lacking the sensor domain is not responsible for UPR signal transduction

Because *ire1a/c* mutants retain fertility and IRE1-dependent UPR signal transduction, *IRE1B* possibly plays a role in both UPR and developmental processes. To investigate an unknown IRE1 function, we generated a construct expressing FLAG-tagged wild-type (WT) form of IRE1B as well as those with kinase (K487A), RNase (K821A), and luminal sensor deletion (ΔLD) mutants under the control of its native promoter (Figs 3A and S2B). For comparison, we generated constructs expressing FLAG-tagged wild-type (WT) IRE1A as well as mutant IRE1A with kinase (K442A) and RNase (K781A) mutation under the *IRE1A* native promoter (Figs 3A and S2B). These constructs were introduced into the *ire1a/b* mutant using *Agrobacterium*-mediated transformation and T$_3$ homozygous transgenic plants were used for further analyses. All of the WT and mutant IRE1 constructs expressed expected sizes of IRE1 proteins in seedlings (Fig 3B). Up- and down-regulation of *BiP3* and *PR-4* mRNA, respectively, by Tm treatment were restored in the FLAG-IRE1A(WT) and FLAG-IRE1B(WT) transgenic plants, but not in kinase-, RNase-, and ΔLD-expressing transgenic plants (Fig 3C). We next analyzed in vivo phosphorylation of FLAG-IRE1B under ER stress by Phos-tag–based Western blot (Yang et al, 2010). A slower migrating, phosphorylated form of IRE1B was detected in Tm- and DTT-treated FLAG-IRE1B(WT) plants but not in the DTT-treated K487A plants (Fig 3D). Consistent with the results of *BiP3* and *PR-4* expressions, expression of FLAG-IRE1B(WT) restored hypersensitivity of the *ire1a/b* mutant to DTT and Tm to the level observed in WT, but expression of ΔLD did not (Fig 3E). These results indicate that the mutant IRE1B lacking the sensor domain does not contribute to the ER stress response.

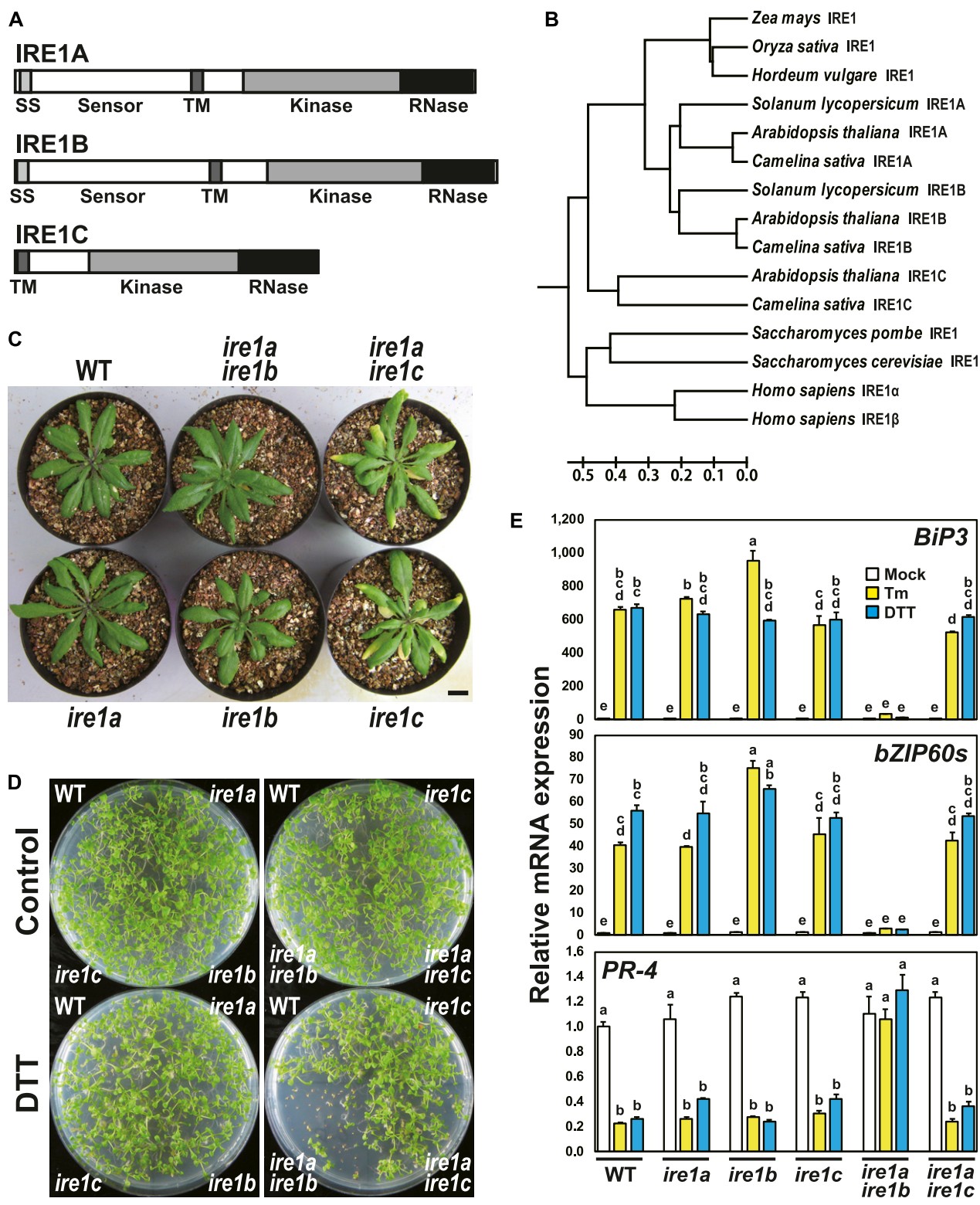

**Figure 1. Arabidopsis IRE1C does not contribute to the UPR.**
**(A)** Structures of Arabidopsis IRE1A, IRE1B, and IRE1C proteins. **(B)** Phylogenetic tree of IRE1 proteins from mammals (*Homo sapiens*), fungi (*Saccharomyces cerevisiae* and *Saccharomyces pombe*) and plants was constructed by UPGMA using MEGA 6 (Tamura et al, 2013). The tree is drawn to scale, with branch lengths in the same units as those of the evolutionary distances used to infer the phylogenetic tree. The evolutionary distances were computed using the Poisson correction method and are in the units of the number of amino acid substitutions per site. **(C)** Wild-type (WT) and *ire1* mutant plants 40 d after germination (DAG). Bar = 10 mm. **(D)** DTT sensitivity of the *ire1* mutants. Seedlings at 15 DAG of the indicated lines were treated with or without 1 mM DTT. **(E)** The relative mRNA levels of *BiP3* (upper), *bZIP60s* (middle), and

**Table 1. Transmission of the *ire1c* allele through the male and female gametophyte in the progenies of the *ire1a ire1b ire1c/+* mutants crossed with the *ire1a ire1b* mutant or self-pollination.**

| Parental Genotype | | Genotypes of Progeny | | | | Observed Ratio | Expected Ratio |
|---|---|---|---|---|---|---|---|
| Female | Male | +/+ | c/+ | c/c | Total | +/+:c/+:c/c | +/+:c/+:c/c |
| a/a b/b c/+ | a/a b/b c/+ | 83 | 25 | 0 | 108 | 1.0:0.30:0[a] | 1:2:1 |
| a/a b/b c/+ | a/a b/b +/+ | 186 | 39 | 0 | 225 | 1.0:0.21:0[a] | 1:1:0 |
| a/a b/b +/+ | a/a b/b c/+ | 119 | 0 | 0 | 119 | 119:0:0[a] | 1:1:0 |

[a]Significantly different from the Mendelian segregation ratio ($\chi^2$, $P < 0.01$).
+, wild-type allele; a, *ire1a* allele; b, *ire1b* allele; c, *ire1c* allele.

### FLAG-IRE1B(WT) and ΔLD restore the developmental defects in *ire1a/b ire1c/+* mutant

We crossed the *ire1a/b* mutant plants expressing FLAG-IRE1B(WT) or ΔLD with *ire1a/b ire1c/+* mutant plants. $F_1$ plants heterozygous for *IRE1C* (*ire1a/b ire1c/+*) were selected and self-pollinated. Among $F_2$ plants, those that are homozygous for the transgenes and heterozygous for

**Figure 2. *ire1a ire1b* homozygous and *ire1c* heterozygous plants show developmental defects.**
**(A)** Self-pollinated progenies of *ire1a ire1b ire1c/+* mutant plants at 40 DAG. Genotypes are shown on the left. Bar = 10 mm. **(B)** Siliques of *ire1a ire1b ire1c/+* (left) and *ire1a ire1b +/+* (right) plants. Bar = 10 mm. **(C)** Reproductive development of *ire1* mutants. Flowers at stage 14 (Smyth et al, 1990; upper; bar = 1 mm), siliques (middle; bar = 3 mm), and anthers at stage 12 stained with Alexander's stain (lower; bar = 100 μm) from wild-type (WT) and *ire1* mutants are shown.

*IRE1C* were selected for further analyses. Growth defects (Fig 4A–F) and the reduction of seed set (Fig 4G–I) in the *ire1a/b ire1c/+* mutant were restored by expression of FLAG-IRE1B(WT) and ΔLD. Abortion of pollen development in *ire1a/b ire1c/+* mutant was restored, as well (Fig 4J–L). At the completion of meiosis, the *ire1a/b ire1c/+* mutant plants expressing FLAG-IRE1B(WT) or ΔLD produced four viable microspores in each tetrad (Fig 4M and N), whereas the *ire1a/b ire1c/+* mutant frequently produced abnormal tetrads (Fig 4O).

### Impaired pollen development in *ire1a/b/c* gametophyte is restored by ΔLD

Unexpectedly, self-pollinated progenies of the *ire1a/b ire1c/+* plants expressing FLAG-IRE1B(WT) or ΔLD segregated with the *IRE1C* allele in ratios of 1.0:1.9:0 (+/+:c/+:c/c; n = 154) and 1.0:1.4:0 (n = 237), respectively (Table 2). Nevertheless, their occurrence ratios of the heterozygous allele were higher than that in the *ire1a/b ire1c/+* mutant (1.0:0.30:0; n = 108; Table 1). To determine whether the impaired transmission of the *ire1c* allele through male gametophyte in *ire1a/b ire1c/+* mutant was restored by FLAG-IRE1B(WT) and ΔLD, we performed reciprocal crosses between the *ire1a/b ire1c/+* plants with homozygous FLAG-IRE1B(WT) or ΔLD transgene and wild-type plants. Consistent with the results of the reciprocal crossing between *ire1a/b* and *ire1a/b ire1c/+* (Table 1), control reciprocal crossing between wild-type and *ire1a/b ire1c/+* showed no transmission of the *ire1c* allele through male gametophyte (Table 2). In the case of the crossing between the wild-type plants as female parents and the *ire1a/b ire1c/+* plants with FLAG-IRE1B(WT) or ΔLD as male parents, progenies having *ire1c/+* allele were obtained in the ratios of 1.0:0.33 (+/+:c/+; n = 57) and 1.0:0.57 (n = 94), respectively (Table 2). These results indicate that not only FLAG-IRE1B(WT) but also ΔLD can compensate for impaired male gametogenesis in the *ire1a/b/c* haplotype.

To investigate the defect of male gametogenesis in the *ire1a/b/c* haplotype, we observed cross sections prepared from inflorescences of wild-type, *ire1a/b ire1c/+*, and *ire1a/b ire1c/+* expressing ΔLD at different developmental stages. The anther size and the number of pollen grains were reduced in *ire1a/b ire1c/+* compared with wild-type (Fig 5). Regarding pollen development, no obvious differences were observed between *ire1a/b ire1c/+* and wild-type at stage 8 and 9 (Fig 5). However, a part of pollen grains collapsed in *ire1a/b ire1c/+* at

*PR-4* (lower) in WT and *ire1* mutants. RNA from seedlings at 10 DAG were treated with 5 mg/l Tm, 2 mM DTT, or mock for 5 h and subjected to qPCR. Data are means ± SEM of three independent experiments. Different letters within each treatment indicate significant differences ($P < 0.05$) by the Tukey–Kramer Honestly Significant Difference (HSD) test. SS, signal sequence; TM, transmembrane domain.

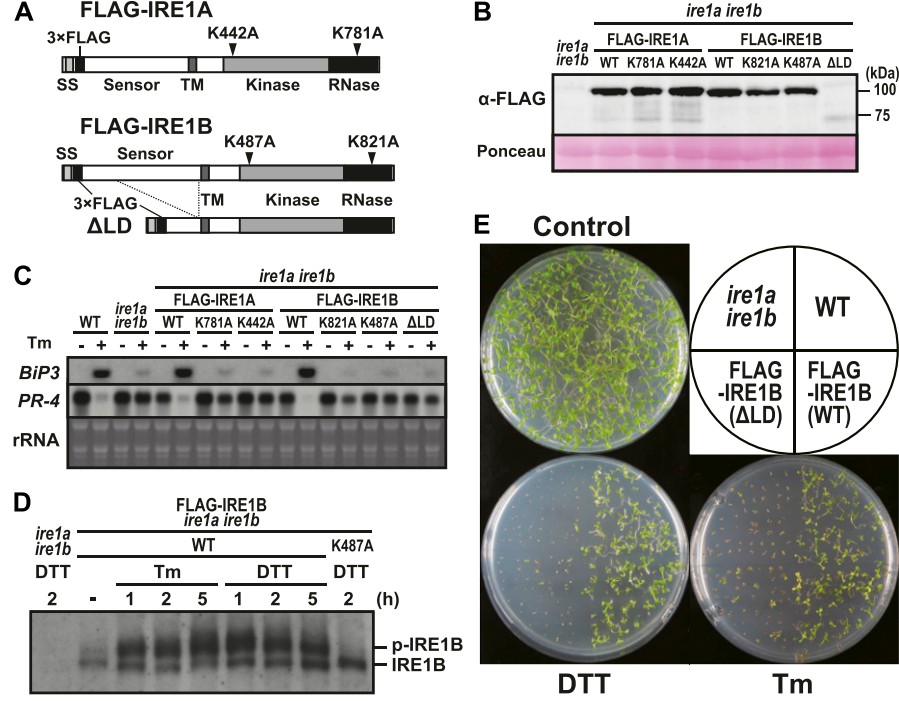

**Figure 3. Transgenic *ire1a ire1b* plants expressing FLAG-tagged wild-type or mutant IRE1.**
**(A)** Schema of FLAG-tagged IRE1 proteins. Mutations of kinase and RNase domains are shown as arrowheads. **(B)** Detection of FLAG-IRE1 proteins in the transgenic *ire1a ire1b* plants with anti-FLAG antibody. Ponceau S staining was used as loading control. **(C)** RNA blot analysis of *BiP3* and *PR-4* in wild-type (WT), *ire1a ire1b* mutant, and transgenic *ire1a ire1b* plants. Seedlings at 10 DAG were treated with (+) or without (−) 5 mg/l Tm for 5 h. **(D)** Detection of FLAG-IRE1B(WT) and FLAG-IRE1B(K487A) with anti-FLAG antibody in the transgenic *ire1a ire1b* plants treated with Tm or DTT. Samples were resolved on Phos-tag SDS–PAGE to detect the phosphorylated FLAG-IRE1B (p-IRE1B). **(E)** DTT and Tm sensitivities of the transgenic *ire1a ire1b* plants. Seedlings at 15 DAG of the indicated lines were treated with or without 1 mM DTT or 0.1 mg/l Tm. SS, signal sequence; TM, transmembrane domain.

stage 11 (Fig 5; indicated by arrowheads). The collapsing pollen grains were also shown in *ire1a/b ire1c/+* expressing ΔLD, but the frequency was very low (Fig 5; arrowhead). In *ire1a/b ire1c/+* expressing ΔLD, the anther size was restored to the wild-type level (Fig 5).

Results of the crossing between *ire1a/b ire1c/+* as female parents and *ire1a/b* (Table 1) or wild-type (Table 2) as male parents suggest incomplete female gametogenesis in the *ire1a/b/c* haplotype. However, *FLAG-IRE1B(WT)* and ΔLD transgenes did not affect the occurrence ratios of heterogeneous *ire1c/+* allele through the female gametophyte (Table 2).

### Different IRE1 activation states by saturated fatty acids in the presence or absence of sensor domain

Growing evidence suggests that yeast and metazoan IRE1 have lipid-dependent activation machinery (Volmer & Ron, 2015). Because exogenous application of glycerol is known to reduce oleic acid (18:1) level in Arabidopsis (Kachroo et al, 2004), we applied glycerol treatment to Arabidopsis seedlings to increase saturated fatty acid composition. As expected, levels of palmitic acid (16:0) and stearic acid (18:0) were increased after 3 d of glycerol treatment in wild-type and *ire1a/b* seedlings (Fig 6A). Glycerol treatment induced *bZIP60* splicing in wild-type but not in *ire1a/b* seedlings (Fig 6B). Impaired *bZIP60* splicing in *ire1a/b* was restored by expression of FLAG-IRE1A(WT) and FLAG-IRE1B(WT), but not by that of kinase, RNase, and ΔLD mutants (Figs 6C and S4A and B). Accumulation and phosphorylation of the FLAG-IRE1B(WT) protein was observed in the glycerol-treated seedlings (Fig 6D). Thus, IRE1's kinase, RNase, and sensor domains are responsible for *bZIP60* splicing under glycerol treatment in vivo.

To determine whether glycerol treatment induces RIDD, mRNA levels of three RIDD target genes (*PR-4*, *PRX34*, and *MBL1*; Mishiba et

al, 2013; Iwata et al, 2016) were analyzed in the glycerol-treated wild-type and *ire1a/b* seedlings, which were further treated with cordycepin to prevent transcription. Higher expressions of the three genes were observed in the glycerol-treated wild-type and *ire1a/b* plants compared with untreated control (Fig 6E). Decrease in *PR-4*, *PRX34*, and *MBL1* mRNA abundance was detected within 5 h of cordycepin treatment in wild-type but not in *ire1a/b* seedlings. The impaired mRNA degradation in *ire1a/b* was restored by expression of FLAG-IRE1B(WT) and ΔLD, but not by that of RNase (K821A) mutant (Fig 6F). Consistently, although levels of mRNA encoding cytosolic proteins, cFBPase and UGPase, did not show significant difference among the samples irrespective of glycerol treatment (Fig S4C), expression of *PR-4*, *PRX34*, *MBL1*, and *PME41* (RIDD target; Mishiba et al, 2013) mRNAs was increased ($P < 0.05$) in *ire1a/b* and K821A compared with wild-type, FLAG-IRE1B(WT), and ΔLD plants under glycerol treatment (Fig S4D). Accumulation of ΔLD proteins was observed under glycerol and DTT treatments (Fig 6G). Phos-tag Western blot of ΔLD protein showed two slower migrating bands than unphosphorylated protein in the untreated-, Tm-, and DTT-treated plants, whereas only the slowest migrating band was detected in the glycerol-treated plants (Fig 6H). The *FLAG-IRE1B(WT)* and ΔLD mRNAs were slightly increased by glycerol treatment (Fig 6I). These results indicate that glycerol treatment stimulate the mutant IRE1 proteins lacking the sensor domain, causing RIDD.

### CRISPR/Cas9-induced deletion corresponding to *IRE1B* sensor region in *ire1a/c* mutant

To gain insight into the contribution of the sensor domain-independent IRE1 activation to the developmental process, we tried to induce deletion in the sensor domain–coding region of the

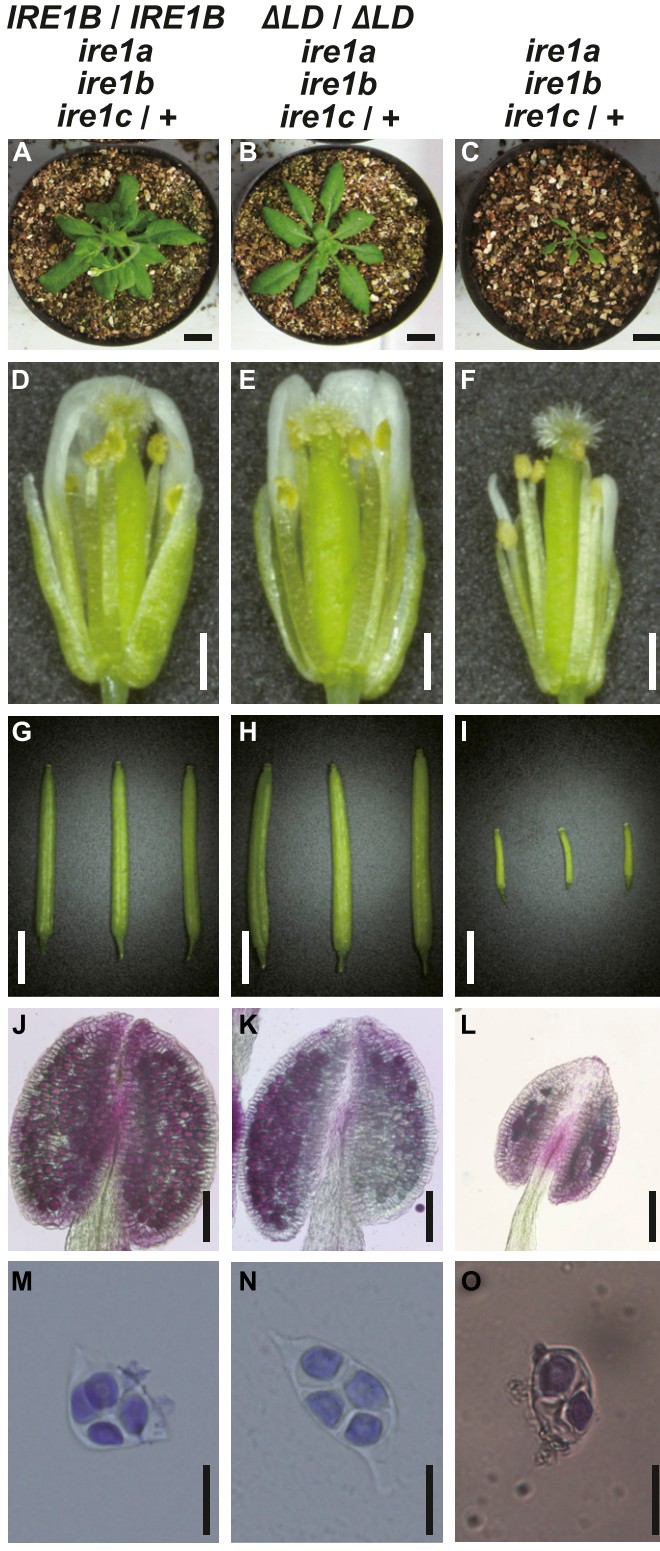

**IRE1B / IRE1B**
**ire1a**
**ire1b**
**ire1c / +**

**ΔLD / ΔLD**
**ire1a**
**ire1b**
**ire1c / +**

**ire1a**
**ire1b**
**ire1c / +**

**Figure 4. Phenotypic complementation of *ire1a ire1b ire1c/+* mutants by *FLAG-IRE1B(WT)* or *ΔLD*.**
Phenotypes of the transgenic *ire1a ire1b ire1c/+* plant having *FLAG-IRE1B(WT)* (left) or *ΔLD* (center), and *ire1a ire1b ire1c/+* plant (right). **(A, B, C)** Plants at 40 DAG. Bar = 10 mm. **(D, E, F)** Flowers at stage 14. Bar = 500 µm. **(G, H, I)** Siliques. Bar = 3 mm. **(J, K, L)** Anthers at stage 12 stained with Alexander's stain. Bar = 100 µm. **(M, N, O)** Tetrads stained with toluidine blue. Bar = 20 µm.

*IRE1B* gene in the *ire1a/c* mutant using CRISPR/Cas9 system. *Agrobacterium* harbouring pKIR1.0 binary vector (Tsutsui & Higashiyama, 2017) containing two gRNAs targeting the 5'- and -3' ends of the IRE1B's sensor domain–coding region (Figs 7A and S2C) was used to transform the *ire1a/c* mutant. We selected T$_2$ lines showing 3:1 segregation ratio for the presence and absence of RFP fluorescence (see Fig S2C) in seeds and picked up some seeds with no fluorescence, indicative of no T-DNA insertion for further analyses (Tsutsui & Higashiyama, 2017). When we amplify an *IRE1B*-coding genomic region by PCR, we obtained smaller bands than that would be expected from intact *IRE1B* in some T$_2$ plants, indicating that deletion is successfully introduced. Therefore, their self-progenies (T$_3$) were used for further analyses. In lines #2–5 and #9–6, all T$_3$ plants analyzed showed a single, smaller *IRE1B* fragment, indicating homozygous deletion of *IRE1B* locus (Fig S5A). Sequence analysis showed deletion of 981- and 1,216-bp regions, each corresponding to part of the sensor domain in #2–5 and #9–6, respectively, and 1-bp deletion at the gRNA2 target site was also detected in #2–5 (Fig 7A). Consistently, RT-PCR showed expression of shorter *IRE1B* mRNA in #2–5 and #9–6 seedlings (Fig S5B). We expected that two isolated lines express N-terminally truncated IRE1B proteins by illegitimate translation (Makino et al, 2016) because translation from the original ATG produces premature N-terminal peptides (52 and 76 aa, respectively). These predicted ORFs in #2–5 and #9–6 have a truncated and an intact transmembrane domain, respectively (Fig 7A). High sensitivity to Tm and DTT equivalent to *ire1a/b* was found in #2–5 and #9–6 lines compared with that in *ire1a/c* (Figs 7B and S5C). Like the *ire1a/b* mutant, up- and down-regulation of *BiP3* and *PR-4* mRNA, respectively, by Tm treatment were diminished in #2–5 and #9–6 lines (Fig 7C). Glycerol treatment–dependent *bZIP60* splicing as shown in *ire1a/c* was also diminished in #2–5 and #9–6 lines (Fig 7D). Nevertheless, mRNA expression of RIDD target genes in these lines was decreased as compared with those in *ire1a/b* under glycerol treatment (Figs 7E and S6A). Growth defects and the reduction in seed set, which occurred in *ire1a/b ire1c/+* mutant, were not observed in the #2–5 and #9–6 plants (Fig S6B and C).

## Discussion

IRE1 is known as the most conserved and sole UPR signal transducer in lower eukaryotes (Mori, 2009). Evolution of multicellular organisms adapt IRE1 functions not only to environmental conditions but also to developmental conditions, as in the fact that IRE1 deficiency causes embryonic lethality in some organisms. In developmental processes, specific cells producing a large amount of secretory proteins, such as β-cells of pancreas (Lee et al, 2011), goblet cells (Tsuru et al, 2013), and dendritic cells (Osorio et al, 2014), activate IRE1 in normal condition. These findings raise a question of whether production of unfolded proteins is prerequisite for the IRE1 activation in these specific cells. The present study showed that IRE1 activation without sensing unfolded protein is required for multifaceted developmental processes in Arabidopsis. We speculate that unfolded protein-independent IRE1 activation is a feature of anticipatory UPR (Vitale & Boston, 2008; Rutkowski & Hegde, 2010) to avoid producing "unprofitable"

**Table 2. Transmission of the *ire1c* allele through the male and female gametophyte in the progenies of the *ire1a ire1b ire1c/+* mutants having *IRE1B* or *ΔLD* transgenes crossed with wild-type or self-pollination.**

| Parental Genotype | | Genotypes of Progeny | | | | Observed Ratio | Expected Ratio |
|---|---|---|---|---|---|---|---|
| Female | Male | +/+ | c/+ | c/c | Total | +/+:c/+:c/c | +/+:c/+:c/c |
| +/+ +/+ +/+ | a/a b/b c/+ | 202 | 0 | 0 | 202 | 202:0:0[a] | 1:1:0 |
| +/+ +/+ +/+ | a/a b/b c/+ IRE1B | 43 | 14 | 0 | 57 | 1.0:0.33:0[a] | 1:1:0 |
| +/+ +/+ +/+ | a/a b/b c/+ ΔLD | 60 | 34 | 0 | 94 | 1.0:0.57:0[a] | 1:1:0 |
| a/a b/b c/+ | +/+ +/+ +/+ | 266 | 81 | 0 | 347 | 1.0:0.30:0[a] | 1:1:0 |
| a/a b/b c/+ IRE1B | +/+ +/+ +/+ | 130 | 32 | 0 | 162 | 1.0:0.25:0[a] | 1:1:0 |
| a/a b/b c/+ ΔLD | +/+ +/+ +/+ | 135 | 25 | 0 | 160 | 1.0:0.19:0[a] | 1:1:0 |
| a/a b/b c/+ IRE1B | a/a b/b c/+ IRE1B | 53 | 101 | 0 | 154 | 1.0:1.9:0[a] | 1:2:1 |
| a/a b/b c/+ ΔLD | a/a b/b c/+ ΔLD | 97 | 140 | 0 | 237 | 1.0:1.4:0[a] | 1:2:1 |

[a]Significantly different from the Mendelian segregation ratio ($\chi^2$, $P < 0.01$).
+, wild-type allele; *a*, *ire1a* allele; *b*, *ire1b* allele; *c*, *ire1c* allele; *IRE1B*, FLAG-IRE1B(WT) transgene (homozygote); *ΔLD*, *ΔLD* transgene (homozygote).

unfolded proteins, as a primer for UPR, during the evolution of an unfolded protein-sensing system in multicellular organisms.

We found that plants heterozygous for the *IRE1C* allele (*ire1c/+*) in *ire1a/b* mutant background display developmental defects of male (and also probably female) gametogenesis, incomplete floral organ formation, and retardation of vegetative growth (Figs 2 and S3D). The incomplete dominance of the *IRE1C* allele is probably due to low expression of the *IRE1C* transcript (Fig S3C and E). *IRE1C* expression is strongest in the anther (Fig S3A) and embryo (Fig S3B), which is somewhat similar to that of *IRE1B* (Koizumi et al, 2001). Expression of *IRE1A* alone may be insufficient for proper Arabidopsis development because we could not obtain *ire1b ire1c* double mutant. Functions and physiological significance of the Arabidopsis IRE1C is still unclear. A closely related species, *Arabidopsis lyrata*, has four *IRE1* genes, *IRE1A* (XP_002884063), *IRE1B* (EFH48369), *IRE1C* (XP_002884871), and *IRE1C-like* (EFH64123). Amino acid sequence similarities between those in *Arabidopsis thaliana* and *A. lyrata* are high in IRE1A (93%)

and IRE1B (91%), whereas IRE1C proteins are more diversified (66%). *A. lyrata* also has *IRE1C-like* gene with low sequence similarity. These results suggest that these *IRE1C* genes probably arose through gene duplication during evolution of Brassicaceae species.

Pollen is known to be particularly sensitive to environmental conditions that disturb protein homeostasis (Fragkostefanakis et al, 2016). In high temperature, *ire1a/b* mutant displays male sterility (Deng et al, 2016), suggesting that pollen development is sensitive to heat stress and that IRE1-dependent UPR pathway is required for protecting male fertility from heat stress. This feature seems to be distinct from the requirement of the UPR-independent IRE1 activation for pollen development in unstressed condition observed in the present study. IRE1 activation in pollen without stress conditions was suggested by detection of bZIP60s in anther (Iwata et al, 2008), whereas *ire1a/b* mutant does not compromise pollen development under normal conditions (Deng et al, 2013, 2016). In the present study, we demonstrate that IRE1C, which lacks a sensor domain, acts redundantly with IRE1A and IRE1B in pollen development. Observation of the pollen development in *ire1a/b ire1c/+* mutant showed reduced number of pollen (Fig 5) and abnormal tetrad (Fig 4O), suggesting that male gametogenesis in the *ire1a/b/c* haplotype is defective in meiosis. In addition, the *ire1a/b ire1c/+* mutant showed collapsed pollen grains at stage 11 (Fig 5), which is somewhat similar to that observed in RNAi-mediated suppression of ER- and Golgi-located phospholipase A₂ transgenic plants (Kim et al, 2011). Together with the results that no transmission of *ire1c* allele was found through *ire1a/b* male gametophyte and that *ΔLD* can restore the transmission in the *ire1a/b/c* haplotype (Tables 1 and 2), the unfolded protein-independent IRE1 activation is required for the male gametogenesis in unstressed conditions. Given the fact that lipids (Piffanelli et al, 1997) and proteins (Mascarenhas, 1975; Holmes-Davis et al, 2005) are largely produced during pollen development, Arabidopsis IRE1 may sense and maintain protein-to-lipid ratio in cellular membranes as suggested in other organisms (Covino et al, 2018; Michell, 2018).

Genetic analysis also showed distorted segregation ratios in progenies of crosses between *ire1a/b ire1c/+* females and *ire1a/b* or wild-type males (Tables 1 and 2), suggesting that the unfolded protein-independent IRE1 activation may be involved in embryogenesis or

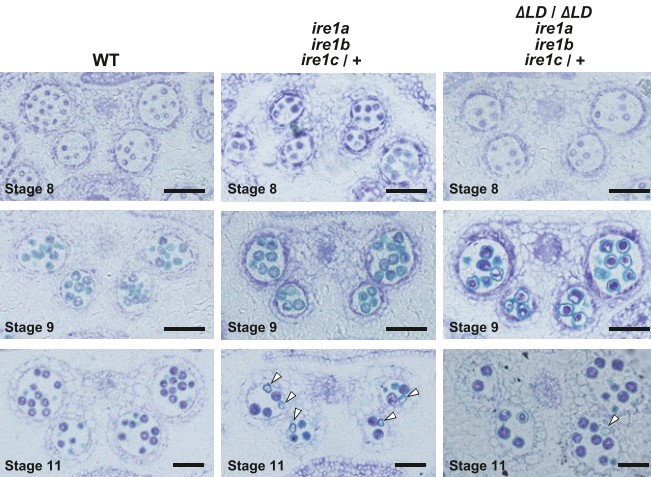

**Figure 5. Abnormal pollen development in *ire1a ire1b ire1c/+* is partially complemented by *ΔLD*.**
Transverse sections of developing anthers at stages 8, 9, and 11 in WT (left), *ire1a ire1b ire1c/+* mutant (middle), and transgenic *ire1a ire1b ire1c/+* plants having *ΔLD* (right). Arrowheads indicate collapsed pollen grains. Bar = 50 μm.

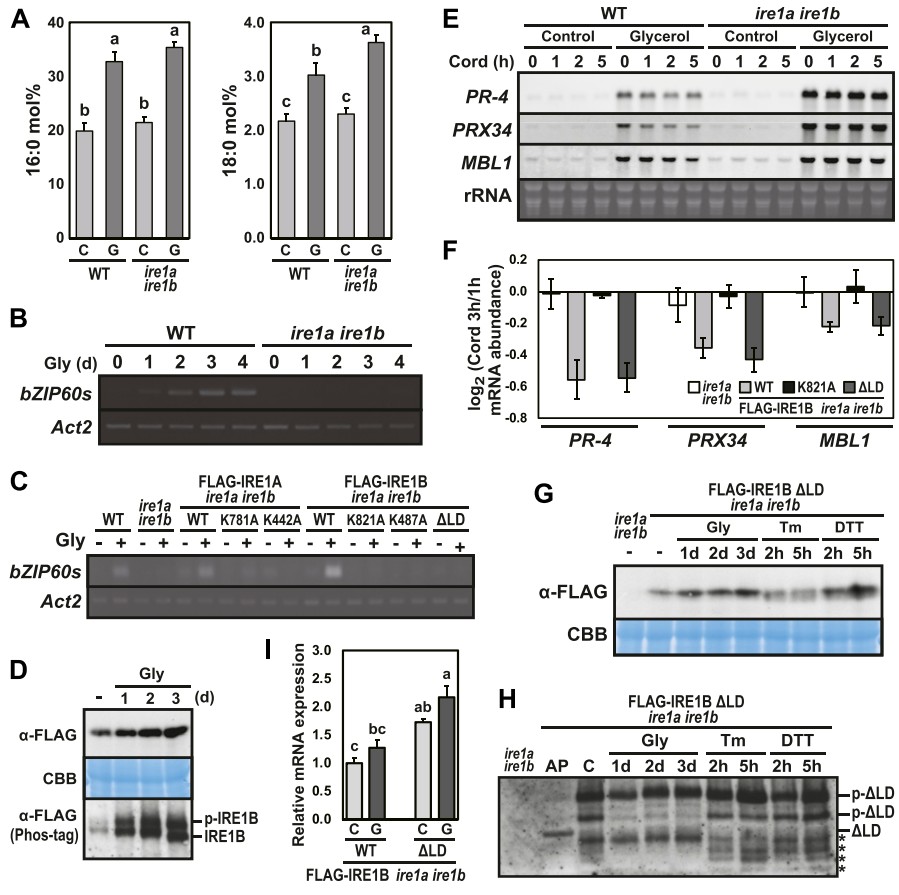

**Figure 6. Glycerol treatment stimulates IRE1 kinase and RNase activities.**
(A) Percentages of saturated fatty acids (16:0 and 18:0) in WT and *ire1a ire1b* plants at 10 DAG treated with (G) or without (C) glycerol for 3 d. Error bars represent SD (n = 6). Different letters within each treatment indicate significant differences (*P* < 0.01) by the Tukey–Kramer HSD test. **(B, C)** Detection of *bZIP60* mRNA splicing in WT, *ire1a ire1b* plants (B), and FLAG-IRE1 transgenic *ire1a ire1b* plants (C) at 10 DAG. RT-PCR was performed using *bZIP60s*-specific primers. *Actin2* (*Act2*) was used as an internal control. **(B)** Glycerol treatment was performed for 0–4 d. **(C)** Plants were treated with (+) or without (−) glycerol for 3 d. **(D)** Detection of FLAG-IRE1B(WT) with anti-FLAG antibody in the transgenic *ire1a ire1b* plants treated with glycerol for 0–3 d. Samples were resolved on SDS–PAGE (upper) and Phos-tag SDS–PAGE (lower) followed by immunodetection with anti-FLAG antibody. An equal loading was shown by CBB staining after SDS–PAGE (middle). **(E)** RNA blot analysis of *PR-4*, *PRX34*, and *MBL1* in WT and *ire1a ire1b* plants at 10 DAG. Plants were treated with or without glycerol for 3 d and plants at 10 DAG were treated with cordycepin (Cord) for 0–5 h. **(F)** Relative mRNA levels of *PR-4*, *PRX34*, and *MBL1* in FLAG-IRE1B(WT, K821A, ΔLD) transgenic *ire1a ire1b* plants. 3-d glycerol-treated plants at 10 DAG were treated with Cord for 1 and 3 h and subjected to qPCR. log$_2$ fold change was calculated by dividing average mRNA level of Cord 3 h by that of Cord 1 h. Data are means ± SEM of 3–6 independent experiments. **(G, H)** Detection of ΔLD with anti-FLAG antibody in the transgenic *ire1a ire1b* plants treated with glycerol, Tm, or DTT. **(G)** Samples were resolved on SDS–PAGE followed by immunodetection with anti-FLAG antibody. CBB staining was used as loading control. **(H)** Samples were resolved on Phos-tag SDS–PAGE to detect the phosphorylated ΔLD (p-ΔLD). Asterisks indicate possible degradation products of ΔLD. **(I)** The relative mRNA expression levels of *FLAG-IRE1B(WT, ΔLD)* transgenes. RNA from seedlings at 10 DAG treated with (G) or without (C) glycerol for 3 d was subjected to qPCR. Data are means ± SEM of four independent experiments. Different letters within each treatment indicate significant differences (*P* < 0.05) by the Tukey–Kramer HSD test. AP, alkaline phosphatase-treated sample.

female gametogenesis. This hypothesis is supported by the observations that the sensor domain–lacking *IRE1B* mutant lines (i.e., #2–5 and #9–6) in *ire1a/c* background set seeds normally (Fig S6C) and that *IRE1B* (Koizumi et al, 2001) and *IRE1C* (Fig S3B) express in the ovule and embryo, respectively. Inconsistent with the result of the #2–5 and #9–6 lines, we could not obtain homozygous *ire1c* mutant plants from selfed progenies of the *ire1a/b ire1c/+* plants expressing FLAG-IRE1B(WT) or ΔLD plants. A possible reconciliation could be that the *IRE1B* transgene promoter is not expressed in embryo at a level equivalent to endogenous *IRE1B* because of epigenetic modifications of the promoter or to insufficiency in the length of the transgene promoter.

By co-expression of two gRNAs and Cas9, part of *IRE1B*-coding regions (981 and 1,216 bp) corresponding to its sensor domain was removed from the *ire1a/c* mutant genome in #2–5 and #9–6, respectively (Fig 7A). In these lines, exclusive production of short peptides (52 and 76 aa, respectively) lacking cytoplasmic region translated from the original IRE1B start codon is inconceivable because the *ire1a/b/c* mutant is lethal. We, therefore, suggest that illegitimate translation occurs, which results in the production of sensor domain-lacking IRE1B (Fig 7A). *IRE1B* mRNA may be compatible with illegitimate translation because its 5′UTR contains uORF and *IRE1B* mRNA degradation by premature stop codon

(Garneau et al, 2007) was not observed in #2–5 and #9–6 (Fig S5B). Therefore, it is most conceivable that expression of sensor domain–lacking IRE1B confers normal seed set in #2–5 and #9–6. This unexpected IRE1 translation may also occur in known *ire1* null mutants in other organisms, such as *Caenorhabditis elegans ire1(v33)* mutant (Shen et al, 2001). Whether a shorter TM domain in #2–5 is functional at a transmembrane domain needs to be elucidated. TM domain of IRE1 is known to sense biophysical properties of the ER membrane (Volmer et al, 2013; Halbleib et al, 2017). In Arabidopsis, both IRE1A and IRE1B cause *bZIP60* splicing by glycerol treatment (Fig 6C), although there is no homology between their TM domains. Further studies focusing on the properties of the TM domains should be conducted.

The present study shows distinct modes of IRE1 activation by saturated fatty acid in vivo. Compared with Tm or DTT treatments (Mishiba et al, 2013), the level of IRE1 activation (i.e., fold induction of *bZIP60s* mRNA and fold reduction of *PR-4* mRNA) by glycerol treatment was low. Whereas full-length IRE1B activates both *bZIP60* splicing and RIDD under glycerol treatment, sensor domain–lacking IRE1B activates RIDD but not *bZIP60* splicing (Figs 6, 7, S4, and S6). Biochemical studies in other model systems showed that oligomerization of IRE1 is required for XBP1/HAC1 cleavage but not RIDD (Tam et al, 2014). If plant IRE1 also acts in the same way, sensor

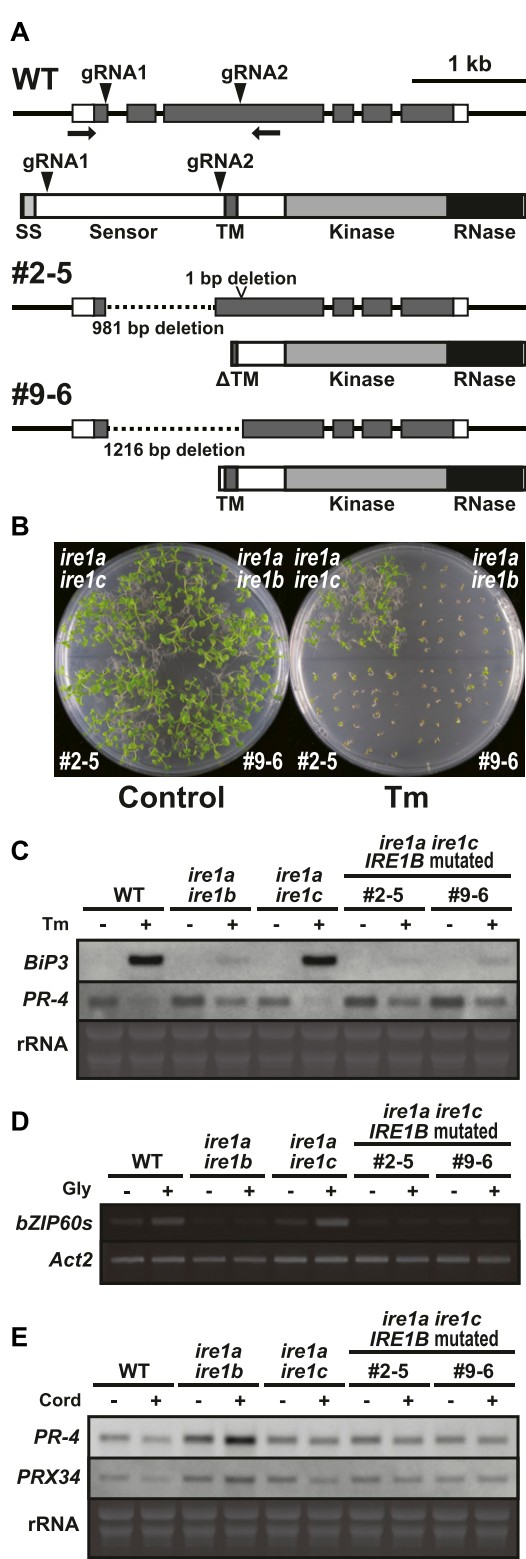

**Figure 7. CRISPR/Cas9-mediated *IREB* gene editing in *ire1a ire1c* mutant.**
**(A)** Schema of hypothetical IRE1B products in the *IRE1B*-mutated *ire1a ire1c* T$_3$ plant lines #2–5 and #1–10. Positions of gRNA target sites are shown as arrowheads. **(B)** Tm sensitivity of the *ire1a ire1b*, *ire1a ire1c*, and *IRE1B*-mutated *ire1a ire1c* plants. Seedlings at 15 DAG of the indicated lines were treated with or without 0.1 mg/l Tm. **(C)** RNA blot analysis of *BiP3* and *PR-4* in WT, *ire1a ire1b*, *ire1a*

domain–lacking IRE1 may be less likely to undergo oligomerization by saturated fatty acid. In vivo phosphorylation of FLAG-IRE1B(WT) was found under glycerol treatment (Fig 6D), whereas phosphorylation of ΔLD is rather complicated. Because ΔLD exhibits multiple phosphorylation states and one of the state is stable regardless of stress (Fig 6H), basal RIDD activity (Maurel et al, 2014) may exist in unstressed tissues. This hypothesis may explain growth retardation of *ire1a/b ire1c/+* plants (Fig 2A) in unstressed conditions. Considering our current observations that *ΔLD* restores developmental defects found in the *ire1a/b ire1c/+* mutant and that deletion of the IRE1B's sensor domain in #2–5 and #9–6 does not prevent their development, RIDD activity may play an important role in the developmental processes. This hypothesis is partially supported by the findings of Deng et al (2013) that *ire1a/b bzip28* but not *bzip60 bzip28* mutant haplotypes impaired male gametogenesis (note that bZIP28 is another UPR arm in Arabidopsis).

In conclusion, this study shows that the unfolded protein-independent IRE1 activation is involved in multifaceted developmental processes, especially pollen development, in Arabidopsis. We hypothesize that the alternative IRE1 activation pathway may be conserved in multicellular organisms as an "anticipatory" mode (Walter & Ron, 2011; Shapiro et al, 2016) of the UPR to avoid producing unfolded proteins in differentiated cells synthesizing a large amount of secretory proteins.

## Materials and Methods

### Plant materials and stress treatments

*A. thaliana* Col-0 ecotype and T-DNA insertion mutants in the Col-0 background were used in this study. Plants were grown on soil or half-strength Murashige and Skoog (1/2 MS) medium containing 0.8% agar and 1% sucrose under 16 h light and 8 h dark conditions at 22°C. T-DNA insertion mutants of *ire1a*, *ire1b*, and *ire1a/b* were described previously (Nagashima et al, 2011). A T-DNA insertion mutant of *IRE1C* (SALK_204405) was obtained from the Arabidopsis Biological Resource Center. T-DNA insertions were confirmed by genomic PCR as shown in Fig S1A and D using primers listed in Table S1. Extraction of DNA for genotyping was carried out as described by Kasajima et al (2004). Genotyping PCR was performed using KAPA Taq Extra PCR kit (Kapa Biosystems) according to the manufacturer's protocol. To test the sensitivity of seedlings to Tm and DTT, sterilized seeds were sown on a 1/2 MS plate containing Tm (0.1 mg/l) or DTT (1 mM). Relative shoot fresh weight of seedlings was calculated as described by Meng et al (2017). For stress treatments, 10-d-old seedlings in 1/2 MS liquid medium (Nagashima et al, 2011)

*ire1c* mutant, and *IRE1B*-mutated *ire1a ire1c* plants. Seedlings at 10 DAG were treated with (+) or without (−) 5 mg/l Tm for 5 h. **(D)** Detection of *bZIP60* mRNA splicing in WT, *ire1a ire1b*, *ire1a ire1c*, and *IRE1B*-mutated *ire1a ire1c* plants at 10 DAG treated with (+) or without (−) glycerol for 3 d. **(E)** RNA blot analysis of *PR-4* and *PRX34* in WT, *ire1a ire1b*, *ire1a ire1c*, and *IRE1B*-mutated *ire1a ire1c* plants at 10 DAG treated with glycerol for 3 d. The samples were treated with (+) or without (−) Cord for 2 h immediately before sampling. SS, signal sequence; TM, transmembrane domain.

were treated with 5 mg/l Tm, 2 mM DTT, or DMSO (mock) for 1–5 h. For glycerol treatment, 7-d-old seedlings in liquid medium were treated with 50 mM glycerol for 3 d followed by treatment with 0.6 mM cordycepin (Wako) for 0–5 h.

## Production of transgenic Arabidopsis plants

For *IRE1C* promoter–*GUS* fusion construct, a 1,577-bp fragment of the *IRE1C* promoter was cloned into pENTR/D-TOPO (Thermo Fisher Scientific) and transferred into pSMAB-GW-GUS (Fig S2A) binary vector by Gateway LR reaction (Thermo Fisher Scientific). For FLAG-tagged IRE1 constructs, 5,126- and 5,081-bp fragments of *IRE1A* and *IRE1B* genes, respectively, comprising ~0.9 and 1.2 kb of their promoter regions, respectively, were amplified by PCR with the primers listed in Table S1 and cloned into pENTR/D-TOPO. Triple FLAG-tag and mutations in the kinase and RNase domains were introduced by PCR. These pENTR vectors were transferred into pSMAB-GW destination binary vector (Fig S2B) by Gateway LR reaction. The ΔLD-expressing vector was made by partial *Mlu*I digestion of pSMAB-FLAG-IRE1B(WT) followed by *Nhe*I digestion, blunt end formation by T4 DNA polymerase, and self-ligation. For CRISPR/Cas9, we used pKIR1.0 binary vector (Tsutsui & Higashiyama, 2017) comprising AtU6 promoter-driven gRNA1 and AtU6 promoter-driven gRNA2 (Fig S2C). The target sequences of the gRNA1 and gRNA2 are listed in Table S1. The binary vectors were introduced into *Rhizobium* strain EHA101 (Hood et al, 1986) and transformed into *ire1a/b* (for *IRE1A* and *IRE1B* constructs), *ire1a/c* (for CRISPR/Cas9) mutants, and wild-type (for *IRE1C* promoter-*GUS*) by the floral dip method (Clough & Bent, 1998).

## RNA analysis

Total RNA was extracted using a NucleoSpin RNA kit (Takara) according to the manufacturer's protocol. For RT-PCR and qPCR, 500 ng of RNA was subjected to RT with random primers using High Capacity cDNA Reverse Transcription Kit (Thermo Fisher Scientific) according to the manufacturer's protocol. qPCR was performed with an ABI 7300 and QuantStudio 3 Real-Time PCR System (Applied Biosystems) using Thunderbird SYBR qPCR Mix (Toyobo), and the transcript abundance of the target genes were normalized to that of 18S rRNA (Zoschke et al, 2007). Primers used for RT-PCR and qPCR are listed in Table S1. RNA gel blot analysis was conducted using DIG High Prime DNA Labeling and Detection Starter Kit II (Roche) according to the manufacturer's protocol. Primers used to generate probes are listed in Table S1.

## Protein analysis

Total protein extraction from Arabidopsis seedlings was performed as described by Liu et al (2011). Protein extracts were fractionated by SDS–PAGE followed by Western blotting with HRP-conjugated anti-FLAG antibody (PM020-7; MBL; 1:10,000) and chemiluminescent detection using Chemi-Lumi One Ultra (Nacalai Tesque). To reveal Rubisco large subunit, Coomassie Brilliant Blue (CBB) staining of the gel or Ponceau-S staining of the membrane were performed. For Phos-tag SDS–PAGE (Kinoshita & Kinoshita-Kikuta, 2011), 6% polyacrylamide gels containing 5–15 $\mu$M Phos-tag acrylamide (Wako) and

10–30 $\mu$M $ZnCl_2$ were run according to the manufacturer's protocol. For Phos-tag SDS–PAGE sample preparation, Arabidopsis seedlings were ground in liquid nitrogen and homogenized in an extraction buffer (100 mM Tris–HCl, pH 7.5, 0.25 M sucrose, and 5 mM PMSF). The homogenate was centrifuged at 2,000$g$ for 2 min (4°C) and the supernatant was centrifuged at 10,000$g$ for 2 min (4°C). The supernatant was further centrifuged at 100,000$g$ for 30 min (4°C). The crude microsomal fraction pellet was subjected to the protein extraction as described above.

## Histological analysis

Anther samples were fixed with 4% glutaraldehyde in 60 mM Hepes (pH 7.0) containing 0.125 M sucrose. After dehydration in a graded series of ethanol/water mixtures, the samples were embedded in Quetol 651 resin (Nisshin EM) with formulation for plant material (Ellis, 2016). Semi-thin (2 $\mu$m) transverse sections were prepared from at least eight resin blocks per sample and stained with 0.2% toluidine blue. Stained sections were examined using a BZ-9000 microscope (Keyence). Histochemical GUS staining of *IRE1C* promoter–*GUS* plants was performed as previously described (Iwata et al, 2008).

## Fatty acid analysis

Arabidopsis seedlings (~150 mg FW) at 10 DAG was used for the analysis of fatty acids, most of which are components of membrane lipids (mainly phospholipids and glycolipids) (Ohlrogge & Browse, 1995). The fatty acids were methylated and extracted using fatty acid methylation kit (Nacalai Tesque) following the manufacturer's instructions. The fatty acid compositions were determined using an Agilent 6890 gas chromatograph (Agilent Technologies) equipped with a DB-23 column (30 m × 0.25 mm × 0.25 $\mu$m; Agilent Technologies). Nonadecanoic methyl ester (C19:0) was used as the internal standard.

# Supplementary Information

# Acknowledgements

We thank Ms Ayumi Sakei, Ms Fumika Yagi, Ms Sae Saito (Osaka Prefecture University), and Dr Yukihiro Nagashima (Texas A&M University) for technical assistance. We also thank the Arabidopsis Biological Resource Center and German plant genomics research program for providing T-DNA insertion lines and Dr Hiroki Tsutsui and Dr Tetsuya Higashiyama (Nagoya University) for pKIR1.0. This work was supported by Grant-in-Aid for Scientific Research (26450010, 17K07610) from the Ministry of Education, Culture, Sports, Science and Technology (MEXT); Shorai Foundation for Science and Technology; and Takeda Science Foundation.

## Author Contributions

K-i Mishiba: conceptualization, resources, data curation, formal analysis, supervision, funding acquisition, validation, investigation,

visualization, methodology, project administration, and writing—original draft, review, and editing.

Y Iwata: resources, formal analysis, investigation, methodology, and writing—original draft, review, and editing.

T Mochizuki: formal analysis, investigation, and visualization.

A Matsumura: formal analysis and investigation.

N Nishioka: resources, formal analysis, validation, and investigation.

R Hirata: formal analysis, validation, and investigation.

N Koizumi: supervision, writing—original draft, and project administration.

## Conflict of Interest Statement

The authors declare that they have no conflict of interest.

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
