## [Reviewer comments · Life Science Alliance]

Life Science Alliance

Unfolded protein-independent IRE1 activation contributes to developmental processes in Arabidopsis

Kei-ichiro Mishiba, Yuji Iwata, Tomofumi Mochizuki, Atsushi Matsumura, Nanami Nishioka, Rikako Hirata, and Nozomu Koizumi

DOI: <https://doi.org/10.26508/lsa.201900459>

Corresponding author(s): Kei-ichiro Mishiba, Osaka Prefecture University

Review Timeline:

Submission Date:	2019-06-18
Editorial Decision:	2019-07-17
Revision Received:	2019-09-13
Editorial Decision:	2019-09-27
Revision Received:	2019-09-29
Accepted:	2019-09-30

Scientific Editor: Andrea Leibfried

Transaction Report:

July 17, 2019

Re: Life Science Alliance manuscript #LSA-2019-00459-T

Dr. Kei-ichiro Mishiba
Osaka Prefecture University
Graduate School of Life and Environmental Sciences
Gakuen 1-1
Nakaku
Sakai, Osaka 599-8531
Japan

Dear Dr. Mishiba,

Thank you for submitting your manuscript entitled "Unfolded protein-independent IRE1 activation contributes to multifaceted developmental processes in Arabidopsis" to Life Science Alliance. The manuscript was assessed by expert reviewers, whose comments are appended to this letter.

As you will see, the reviewers appreciate your analyses and provide constructive input on how to provide better support for some of your conclusions. We would thus like to invite you to submit a revised version to us. The time-demanding complementation assay suggested by ref#2 in point 1 (expression of IRE1A/B::IRE1C in the ire1a ire1b mutant plants exposed to tunicamycin and DTT) is not mandatorily needed for acceptance here, but the other comments should get addressed following the constructive input provided by the reviewers. We would be happy to discuss the individual revision points further with you should this be helpful.

Thank you for this interesting contribution to Life Science Alliance. We are looking forward to receiving your revised manuscript.

Sincerely,

B. MANUSCRIPT ORGANIZATION AND FORMATTING:

Reviewer #1 (Comments to the Authors (Required)):

Review on the manuscript ,Unfolded protein-independent IRE1 activation contributes to

multifaceted developmental processes in Arabidopsis' by Kei-ichiro Mishiba et al.

The authors study the unfolded proteins response in Arabidopsis mediated by the inositol requiring enzyme (IRE1). Two genes, IRE1A and IRE1B, encode for UPR-transducers with an ER-luminal sensor domain for unfolded proteins, while a third gene, which studied in this manuscript and designated as IRE1C, lacks such a sensor domain. The effector functions of IRE1 include an unconventional splicing of bZIP60 mRNA (encoding a transcription factor) and regulated IRE1-dependent decay (RIDD) of mRNAs.

The authors show that a triple mutant lacking IRE1A, IRE1B and IRE1C is not viable, and that a heterozygous IRE1C (*ire1c/+*) in an *ire1a/b* mutant background shows severe growth defects and a reduction of the number of pollen.

The authors find that a genetically-engineered variant of IRE1B lacking its ER-luminal sensor domain for unfolded proteins (*deltaLD*) does not complement for the defects in cytoplasmic splicing of the bZIP60 mRNA and RIDD in a *ire1a/b* double mutant background. However, this *deltaLD* variant complements a developmental defect in the male gametophyte in *ire1 a/b/c* haplotype. The findings suggest a role of IRE1 in plant development and gametogenesis. Furthermore, these findings suggest that signals other than ER-luminal unfolded proteins can activate IRE1 in Arabidopsis.

The study is well-structured, the data are well presented, and the conclusions are supported by the data. The manuscript is good material for the broad readership of Life Science Alliance given that the following points can be addressed.

Major point 1: A possible concern with the complementation experiments using either IRE1B WT, K821A, or the *deltaLD* mutant (Figure 3D) is that the expression level of the *deltaLD* variant seems much higher than that of the WT one (as suggested by Figure 3B). The authors should provide a quantification of the relative expression levels. If possible, in a reasonable amount of time, the authors should perform additional experiments to test if the cytosolic domain of IRE1B is still capable to activate RIDD in glycerol-treated plants (as studied in Figure 6F), when it attached to a different transmembrane domain. This is important, because the transmembrane domain of IRE1 acts as a sensor for membrane-based signals. Furthermore, the authors should determine the expression level of FLAG-IRE1B WT and FLAG-IRE1B *deltaLD* in glycerol-treated plants.

Major point 2: I do not understand the logic of the hexane extraction for the lipid analysis. Hexane is an apolar solvent, which does not efficiently extract membrane lipids. Instead, it extracts free fatty acids (which are also analyzed in the manuscript). However, the most prominently discussed models related the protein-independent activation of the UPR (PMID: 25543896; PMID: 29787971) suggest that the sensing relies on saturated membrane lipids (and not just free fatty acids). The authors should explain their rationale better and/or provide more evidence related to the composition of the membrane lipids of the plant under these conditions.

Major point 1: The authors suggest that a mechanism independent of unfolded proteins mounts a 'pre-emptive' UPR as already previously speculated by others and as referenced in the manuscript. The alternative possibility that the UPR senses a membrane-based signal in order to maintain the protein-to-lipid ratio in cellular membranes as suggested by Mitchell and Ernst laboratories (PMID: 29859544; PMID: 30075144) should be considered and discussed. It is conceivable that the unusual transmembrane domain of IRE1 would localize the protein (and thus RIDD-activity) to specific regions of the ER during stress induced by saturated fatty acids.

Minor point 2: Important references providing evidence for an UPR-activation independently of unfolded ER-luminal proteins should be mentioned (PMID: 22219379 or PMID: 24710536). Furthermore, important references that described the activation of the UPR by saturated lipids should be mentioned (PMID: 19302420; PMID: 20489212).

Minor point 3: The sentence 'It is inconceivable that illegitimate translation that results in sensor domain-lacking IRE1B (Fig. 7A) does not occur, because the *ire1a/b/c* mutant is lethal.' is somewhat hard to read. I would suggest rephrasing it with fewer negations.

Reviewer #2 (Comments to the Authors (Required)):

Summary:

This work by Mishiba et al. provides important advances which not only clarify previously contradictory evidence related to the functional significance of the highly conserved ER stress response master regulator IRE1 during the growth, development, and reproductive fitness of dicot plants, but also advances the field's understanding of IRE1 activation requirements and downstream mechanisms under these conditions in vivo in a whole multicellular organism. The primary narrative of the manuscript focuses on the importance of unfolded-protein independent activation of IRE1 under normal conditions which can complement some of the severe developmental defects of the *ire1a ire1b ire1c/+* genotype. Through an analysis of IRE1B and IRE1B Δ LD complemented *ire1a ire1b* mutants subjected to lipid disequilibrium stress, the authors show overaccumulation of saturated lipids can activate IRE1 and that RIDD functionality in the *ire1a ire1b* can be restored by IRE1B Δ LD but not bZIP60 splicing. The authors suggest the phenotypic rescue of the *ire1a ire1b ire1c/+* mutant by IRE1B Δ LD may be related to the lipid induced RIDD complementation by IRE1B Δ LD. Although lipid stress has been shown to activate mammalian and yeast IRE1 homologs the conservation of this trait in plants had yet to be demonstrated prior to this manuscript. Additionally, the relevance of this sensing function has yet to be connected to developmental programming in multicellular eukaryotes. On the whole, this manuscript was well received and represents a significant contribution to the field, however some conclusions outside of the primary narrative require additional experimental support, or a more nuanced consideration/explanation of select results in order to avoid premature dismissal of alternative possibilities that would require further experimentation which may be outside of the scope of this specific investigation.

Primary Conclusions:

1. IRE1C Loss of function does not alter ER Stress Response (Insufficiently supported).

In this section the authors' qualitative results, a lack of experimental replicates, and the use of different ER stress inducing drug treatments in different experiments leaves the reader with significant room for interpretation. Tunicamycin induced ER stress dependent induction of BIP3 expression is still visible in the *ire1a ire1b* mutant in Figures 1E, 3C, 7C, with a concomitant reduction with PR4 transcript level. The Tm+ BIP3 band in Figure 1E from *ire1a ire1c* also looks slightly lighter than the WT or single mutant Tm+ BIP3 bands. This residual ER stress induced RNase activity in an *ire1a ire1b* mutant (RIDD, and bZIP60 splicing) has been observed in other publications (Ruberti et al. 2018 Figure 5; DOI:10.1111/tbj.13768) and could be caused by IRE1C (assuming a complete ablation of IRE1A and IRE1B RNase activity in the double mutant). This low

level of bZIP60 splicing could be functionally significant, as this previous publication also showed that *ire1a ire1b bzip60* triple mutant seedlings have a more severe ER stress recovery phenotype than the *ire1a ire1b* double mutants (Ruberti et al. 2018 Figure 3 DOI:10.1111/tpj.13768). Assuming multiple independent experiments for Tm treatments in Figure 1E were performed like those in Figure S4 and Figure S6 for glycerol experiments, the authors could use the RNA already produced for Figure 1E to perform qRT-PCR and directly measure the spliced bZIP60 levels quantitatively in WT, *ire1a*, *ire1b*, *ire1c*, *ire1a ire1b*, and *ire1a ire1c* like they did for the glycerol treatment Figure S4. As this is the first figure in the paper, the authors should also unify their molecular and growth phenotype experiments by providing results that use the same drug (i.e. Figure 1D used DTT, while figure 1E uses Tunicamycin) and provide quantitative phenotype data (i.e. shoot fresh weight measurements) to back their observations. Ideally both Tm and DTT would be used in Figure 1D and 1E to show their equivalence. If the necessary RNA samples and seed stocks are available this could be accomplished in ~2 weeks, and would considerably strengthen the alternative conclusion that IRE1C does not likely contribute to bZIP60 splicing and the canonical ER stress response. Given that an ideal phenotypic comparison of the ER stress sensitivity of the *ire1a ire1b*, *ire1a ire1c*, *ire1b ire1c*, *ire1a ire1b ire1c* mutations is not possible due to the lethal nature of the *ire1b ire1c* mutation (as mentioned in the discussion but not supported with data in the results), the authors could fully support their original conclusion that the IRE1C proteins (which only shares ~40% sequence identity with IRE1B transmembrane and luminal domains compared to a 60% shared identity between IRE1A and IRE1B transmembrane and luminal domains) are not activated in an unfolded-protein dependent manner and do not contribute to canonical bZIP60 splicing by complementing *ire1a ire1b* mutants with an IRE1C driven by the IRE1A or IRE1B promoters and subjecting these transgenic lines to Tunicamycin and DTT induced ER stress, like the authors' have done with the FLAG-IRE1B Δ LD lines in Figure 3. However, the time needed to generate these constructs, transform the lines, and reach the appropriate generation for analysis represents many months of work.

2. Mutant IRE1B lacking the sensor domain (FLAG-IRE1B Δ LD) cannot rescue UPR signal transduction (Strongly supported).

Although Figure 3 suffers from a similar technical issue as Figure 1, where different panels use different ER stress inducing drugs, the array of transgenic complementation lines broadly support the authors' conclusion. If the authors were to complete the phenotyping experiment in Figure 3E with Tunicamycin (which was used in the previous panels of figure 3), it would provide further confidence in this conclusion. This additional experiment could also be accomplished in ~2 weeks.

3. A triple mutant of IRE1A IRE1B and IRE1C is lethal, and the *ire1a ire1b ire1c/+* displays growth defects and defective transmission of the *ire1a/b/c* haplotype through pollen are rescued by the FLAG-IRE1B Δ LD (Strongly supported).

Data presented in Tables 1 and 2, Figures 2, 4, 5 properly support the conclusion that an IRE1B variant which cannot sense unfolded proteins(FLAG-IRE1B Δ LD), is able to functionally complement certain growth and reproductive defects similar to the WT IRE1B protein. The authors acknowledge the unusual observation that the selfed *ire1a ire1b ire1c/+* FLAG IRE1B and *ire1a ire1b ire1c/+*FLAG IRE1B Δ LD do not produce *ire1c* homozygote progeny and offer a reasonable alternative explanation with their results in Figure S3 and in the discussion when compared to the Δ LD IRE1B CRISPR/Cas9 mutants #2-5 and #9-6.

It should be noted that the authors observations that no visible pIRE1C:GUS staining was observed in vegetative tissues runs contradictory to the observed growth defect phenotype in the *ire1a ire1b ire1c/+* plants which do not look to be at the reproductive stage. After flowering, IRE1C expression in reproductive tissues could hypothetically lead to systemic signals which could affect vegetative growth, but could the authors offer an alternative explanation in the discussion as to how a plant

without any tissues which express IRE1C, display such dramatic effects from the genetic ablation of one IRE1C allele? Is it conceivable that genetic compensation by increased expression of IRE1C in the *ire1a ire1b* double mutant could be reducing developmental growth defects in the *ire1a ire1b* mutant? To resolve these contradictions and simultaneously demonstrate the transcript knockout status which was demonstrated previously for *ire1a* and *ire1b* in Nagashima et al 2011. but yet to be shown for *ire1c* (Salk_204405) in this manuscript, an expression analysis of IRE1C in WT, *ire1a ire1b*, *ire1c*, and *ire1a ire1b ire1c/+* via qRT-PCR in leaves, shoot meristem, roots, using whole floral tissue (like that in Figure S3A) as a reference point would help. Although not expressly required, this experiment (accomplished in ~ 1 month) could strengthen the overall manuscript without requiring the generation of additional transgenic GUS lines.

4. Different IRE1 Activation states by saturated fatty acids in the presence or absence of the IRE1B sensor domain (Partially supported).

On the whole the conclusion reached by the authors that glycerol treatment induces increases in saturated fatty acid content and concomitant IRE1 activation leading to increases the levels of spliced bZIP60 dependent upon the both the luminal and cytoplasmic functions of IRE1A and IRE1B is fully supported by Figures 6A, 6B, 6C, and most strongly supported by 6SA and 6SB.

However, the authors' assertion that RIDD activity is restored by IRE1B Δ LD, a conclusion central to the overall narrative of the manuscript, could be more strongly supported by adopting a more consistent experimental protocol and presenting more experimental replicates. Discrepancies between results in the semi-quantitative data found in figures 6E, 6F (which do not present any experimental replicates) and the quantitative data from Figure S4D, strongly subtract from reader confidence. Cordycepin, a transcription inhibitor is used to demonstrate that degradation of RIDD targets by IRE1A and IRE1B is an important determinant of the overall transcript level. While the clear decrease of PRX34 transcript levels is observed in figure 6E between 0 and 2 hours, a corresponding decrease in 6F is not readily discernable. PR4 degradation over 0-5 hours of cordycepin treatment is not easily recognized. Importantly, the PR4 and PRX34 0h and 2 hour bands from the FLAG IRE1B Δ LD samples in Figure 6F look much closer in strength to PR4 and PRX34 0h and 2 hour bands from the *ire1a ire1b* samples than FLAG-IRE1B as suggested in the text. A qPCR experiment quantifying transcript levels of PR4 in WT, *ire1a ire1b*, FLAG-IRE1B WT, *ire1a ire1b*, FLAG-IRE1B K821A *ire1a ire1b* and FLAG- IRE1B Δ LD *ire1a ire1b* plants treated with glycerol for 3 days and subsequently treated with cordycepin for 0, 2, and 5 would be sufficient to replace both figures 6E and 6F and strongly support the authors' conclusion.

Additional Comments:

Introduction

Paragraph 1, Line 7: Suggested rewording: Under ER stress IRE1 senses ER luminal unfolded proteins, ultimately leading to IRE1 dimerization, autophosphorylation and RNase activation.

Paragraph 2, Line 6: Suggested rewording: The disparate phenotypic consequences of IRE1 mutation between Arabidopsis and rice...

Paragraph 3, Line 6: Suggested rewording: ...presumed that unfolded protein-independent mechanisms allow cells to preemptively adapt their.....

Discussion

Paragraph 2, Line 4: Suggested rewording:due to low expression of the IRE1C transcript. IRE1C expression is strongest in the anther.....

Reviewer #1 (Comments to the Authors (Required)):

Review on the manuscript 'Unfolded protein-independent IRE1 activation contributes to multifaceted developmental processes in Arabidopsis' by Kei-ichiro Mishiba et al.

The authors study the unfolded proteins response in Arabidopsis mediated by the inositol requiring enzyme (IRE1). Two genes, IRE1A and IRE1B, encode for UPR-transducers with an ER-luminal sensor domain for unfolded proteins, while a third gene, which studied in this manuscript and designated as IRE1C, lacks such a sensor domain. The effector functions of IRE1 include an unconventional splicing of bZIP60 mRNA (encoding a transcription factor) and regulated IRE1-dependent decay (RIDD) of mRNAs.

The authors show that a triple mutant lacking IRE1A, IRE1B and IRE1C is not viable, and that a heterozygous IRE1C (*ire1c/+*) in an *ire1a/b* mutant background shows severe growth defects and a reduction of the number of pollen.

The authors find that a genetically-engineered variant of IRE1B lacking its ER-luminal sensor domain for unfolded proteins (*deltaLD*) does not complement for the defects in cytoplasmic splicing of the bZIP60 mRNA and RIDD in a *ire1a/b* double mutant background. However, this *deltaLD* variant complements a developmental defect in the male gametophyte in *ire1 a/b/c* haplotype. The findings suggest a role of IRE1 in plant development and gametogenesis. Furthermore, these findings suggest that signals other than ER-luminal unfolded proteins can activate IRE1 in Arabidopsis.

The study is well-structured, the data are well presented, and the conclusions are supported by the data. The manuscript is good material for the broad readership of Life Science Alliance given that the following points can be addressed.

Thank you very much for your valuable comments. Our responses and corresponding revisions are presented after your comments (blue letters).

Major point 1: A possible concern with the complementation experiments using either IRE1B WT, K821A, or the deltaLD mutant (Figure 3D) is that the expression level of the deltaLD variant seems much higher than that of the WT one (as suggested by Figure 3B). The authors should provide a quantification of the relative expression levels. If possible, in a reasonable amount of time, the authors should perform additional experiments to test if the cytosolic domain of IRE1B is still capable to activate RIDD in glycerol-treated plants (as studied in Figure 6F), when it attached to a different transmembrane domain. This is important, because the transmembrane domain of IRE1 acts as a sensor for membrane-based signals. Furthermore, the authors should determine the expression level of FLAG-IRE1B WT and FLAG-IRE1B deltaLD in glycerol-treated plants.

To visually compare the sizes between FLAG-IRE1B(WT) and FLAG-IRE1B(Δ LD) proteins, we used a Δ LD line with high expression (may be due to multicopy T-DNA insertion) for western blot in Fig. 3B despite that the line was not used for the complementation analysis. To avoid confusion, we omitted this panel and provided a new blot containing the Δ LD line used for the complementation analysis (Fig. 3B). As the results, expression level of Δ LD protein in the line seemed to be lower than those of other FLAG-IRE1A/B proteins, whereas the Δ LD mRNA expression level was rather (1.7-fold) higher than that of the FLAG-IRE1B(WT) (Fig. 6I). According to your suggestion, we also analyzed the mRNA expression level of FLAG-IRE1B(WT) and Δ LD in glycerol-treated plants by qRT-PCR, which is shown in the additional Fig. 6I. As the result, the transgene mRNA expression level tends to increase after glycerol treatment, while there was no significant difference between the treatment and control. We are so sorry but it is quite difficult to produce and analyze transgenic plants having modified-IRE1B attached to different transmembrane domain within a time frame. Nonetheless, we discussed the possibility whether the TM domain of the Arabidopsis IRE1 senses membrane-based signals (P10, line 16-20).

Major point 2: I do not understand the logic of the hexane extraction for the lipid analysis. Hexane is an apolar solvent, which does not efficiently extract membrane lipids. Instead, it extracts free fatty acids (which are also analyzed in the manuscript). However, the most prominently discussed models related the protein-independent activation of the UPR (PMID: 25543896; PMID: 29787971) suggest that the sensing relies on saturated membrane lipids (and not just free fatty acids). The authors

should explain their rationale better and/or provide more evidence related to the composition of the membrane lipids of the plant under these conditions.

Since the fatty acid methylation kit used in the present study comprises Reagent A (52% toluene and 48% methanol), Reagent B (93% methanol) and Reagent C (30% methanol; detailed chemical compositions of the reagents are unavailable) for preparation of samples before isolation of fatty acids using Isolation Reagent containing 96% hexane, the methylation kit is compatible with extraction of membrane lipids. To avoid confusion, we rewrote the methods for fatty acid analysis. Our results shown in Fig. 6A is, therefore, reflected in the composition of membrane lipids, because most of lipids in plant vegetative cells are found in membranes (PMID: 7640528).

Major point 1: The authors suggest that a mechanism independent of unfolded proteins mounts a 'pre-emptive' UPR as already previously speculated by others and as referenced in the manuscript. The alternative possibility that the UPR senses a membrane-based signal in order to maintain the protein-to-lipid ratio in cellular membranes as suggested by Mitchell and Ernst laboratories (PMID: 29859544; PMID: 30075144) should be considered and discussed. It is conceivable that the unusual transmembrane domain of IRE1 would localize the protein (and thus RIDD-activity) to specific regions of the ER during stress induced by saturated fatty acids.

According to your suggestion, we discussed the alternative possibility (P9, line 26-29) with the suggested references.

Minor point 2: Important references providing evidence for an UPR-activation independently of unfolded ER-luminal proteins should be mentioned (PMID: 22219379 or PMID: 24710536). Furthermore, important references that described the activation of the UPR by saturated lipids should be mentioned (PMID: 19302420; PMID: 20489212).

We added and mentioned these references in Introduction (P3, line 16-19).

Minor point 3: The sentence 'It is inconceivable that illegitimate translation that results in sensor domain-lacking IRE1B (Fig. 7A) does not occur, because the ire1a/b/c mutant is lethal.' is somewhat hard to read. I would suggest rephrasing it with fewer negations.

We rewrote the sentence to avoid double negation (P10, line 6-10).

Reviewer #2 (Comments to the Authors (Required)):

Summary:

This work by Mishiba et al. provides important advances which not only clarify previously contradictory evidence related to the functional significance of the highly conserved ER stress response master regulator IRE1 during the growth, development, and reproductive fitness of dicot plants, but also advances the field's understanding of IRE1 activation requirements and downstream mechanisms under these conditions *in vivo* in a whole multicellular organism. The primary narrative of the manuscript focuses on the importance of unfolded-protein independent activation of IRE1 under normal conditions which can complement some of the severe developmental defects of the *ire1a ire1b ire1c/+* genotype. Through an analysis of IRE1B and IRE1BΔLD complemented *ire1a ire1b* mutants subjected to lipid disequilibrium stress, the authors show overaccumulation of saturated lipids can activate IRE1 and that RIDD functionality in the *ire1a ire1b* can be restored by IRE1BΔLD but not bZIP60 splicing. The authors suggest the phenotypic rescue of the *ire1a ire1b ire1c/+* mutant by IRE1BΔLD may be related to the lipid induced RIDD complementation by IRE1BΔLD. Although lipid stress has been shown to activate mammalian and yeast IRE1 homologs the conservation of this trait in plants had yet to be demonstrated prior to this manuscript. Additionally, the relevance of this sensing function has yet to be connected to developmental programming in multicellular eukaryotes. On the whole, this manuscript was well received and represents a significant contribution to the field, however some conclusions outside of the primary narrative require additional experimental support, or a more nuanced consideration/explanation of select results in order to avoid premature dismissal of alternative possibilities that would require further experimentation which may be outside of the scope of this specific investigation.

We appreciate your review and comments. The points of revision and our replies are described following your comments (blue letters).

Primary Conclusions:

1. IRE1C Loss of function does not alter ER Stress Response (Insufficiently supported).

In this section the authors' qualitative results, a lack of experimental replicates, and the use of different ER stress inducing drug treatments in different experiments leaves the reader with significant room for interpretation. Tunicamycin induced ER stress dependent induction of BIP3 expression is still visible in the *ire1a ire1b* mutant in Figures 1E, 3C, 7C, with a concomitant reduction with PR4 transcript level. The Tm+ BIP3 band in Figure 1E from *ire1a ire1c* also looks slightly lighter than the WT or single mutant Tm+ BIP3 bands. This residual ER stress induced RNase activity in an *ire1a ire1b* mutant (RIDD, and bZIP60 splicing) has been observed in other publications (Ruberti et al. 2018 Figure 5; DOI:10.1111/tpj.13768) and could be caused by IRE1C (assuming a complete ablation of IRE1A and IRE1B RNase activity in the double mutant). This low level of bZIP60 splicing could be functionally significant, as this previous publication also showed

that *ire1a ire1b bzip60* triple mutant seedlings have a more severe ER stress recovery phenotype than the *ire1a ire1b* double mutants (Ruberti et al. 2018 Figure 3 DOI:10.1111/tpj.13768). Assuming multiple independent experiments for Tm treatments in Figure 1E were performed like those in Figure S4 and Figure S6 for glycerol experiments, the authors could use the RNA already produced for Figure 1E to perform qRT-PCR and directly measure the spliced bZIP60 levels quantitatively in WT, *ire1a*, *ire1b*, *ire1c*, *ire1a ire1b*, and *ire1a ire1c* like they did for the glycerol treatment Figure S4.

According to your suggestion, we performed qRT-PCR of spliced *bZIP60*, *Bip3* and *PR-4* in Tm- and DTT-treated WT and *ire1* mutants. The results were shown in Fig. 1E instead of the RNA gel blot panel. The result shows that the *bZIP60* splicing and RIDD activities in *ire1a ire1c* mutant is equivalent to WT. At this time, we cannot completely deny the possibility that fractional *bZIP60* splicing activity detected in the Tm- and DTT-treated *ire1a ire1b* double mutant (Fig. 1E) is caused by IRE1C activation, even though there were no significant differences among the treatments. Given the fact that TM domain of IRE1C is located at the C-terminus (Fig. 1A), it is inconceivable that IRE1C sense the unfolded proteins in the ER lumen.

As this is the first figure in the paper, the authors should also unify their molecular and growth phenotype experiments by providing results that use the same drug (i.e. Figure 1D used DTT, while figure 1E uses Tunicamycin) and provide quantitative phenotype data (i.e. shoot fresh weight measurements) to back their observations. Ideally both Tm and DTT would be used in Figure 1D and 1E to show their equivalence. If the necessary RNA samples and seed stocks are available this could be accomplished in ~2 weeks, and would considerably strengthen the alternative conclusion that IRE1C does not likely contribute to bZIP60 splicing and the canonical ER stress response. Given that an ideal phenotypic comparison of the ER stress sensitivity of the *ire1a ire1b*, *ire1a ire1c*, *ire1b ire1c*, *ire1a ire1b ire1c* mutations is not possible due to the lethal nature of the *ire1b ire1c* mutation (as mentioned in the discussion but not supported with data in the results), the authors could fully support their original conclusion that the IRE1C proteins (which only shares ~40% sequence identity with IRE1B transmembrane and luminal domains compared to a 60% shared identity between IRE1A and IRE1B transmembrane and luminal domains) are not activated in an unfolded-protein dependent manner and do not contribute to canonical bZIP60 splicing by complementing *ire1a ire1b* mutants with an IRE1C driven by the IRE1A or IRE1B promoters and subjecting these transgenic lines to Tunicamycin and DTT induced ER stress, like the authors' have done with the FLAG-IRE1B Δ LD lines in Figure 3. However, the time needed to generate these constructs, transform the lines, and reach the appropriate generation for analysis represents many months of work.

We performed growth phenotype experiments, including shoot fresh weight measurements, under Tm and DTT treatments. The additional data is shown in Figs. S1B and S1C. These results show that

the Tm/DTT responses in *ire1a*, *ire1b*, *ire1c*, and *ire1ac* mutants are equivalent to those in WT. Together with the results of the qRT-PCR described above, we conclude that IRE1C does not likely contribute to the canonical ER stress response. We are so sorry but it is quite difficult to produce and analyze transgenic plants having *IRE1A/B* promoter-driven *IRE1C* transgenes within a time frame.

2. Mutant IRE1B lacking the sensor domain (FLAG-IRE1B Δ LD) cannot rescue UPR signal transduction (Strongly supported).

Although Figure 3 suffers from a similar technical issue as Figure 1, where different panels use different ER stress inducing drugs, the array of transgenic complementation lines broadly support the authors' conclusion. If the authors were to complete the phenotyping experiment in Figure 3E with Tunicamycin (which was used in the previous panels of figure 3), it would provide further confidence in this conclusion. This additional experiment could also be accomplished in ~2 weeks.

According to your suggestion, we provided a picture of Tm-treated plants, in addition to DTT-treated plants, in Fig. 3E. As the result, the Tm treatment gives the same result as the DTT treatment.

3. A triple mutant of IRE1A IRE1B and IRE1C is lethal, and the *ire1a ire1b ire1c/+* displays growth defects and defective transmission of the *ire1a/b/c* haplotype through pollen are rescued by the FLAG-IRE1B Δ LD (Strongly supported).

Data presented in Tables 1 and 2, Figures 2, 4, 5 properly support the conclusion that an IRE1B variant which cannot sense unfolded proteins(FLAG-IRE1B Δ LD), is able to functionally complement certain growth and reproductive defects similar to the WT IRE1B protein. The authors acknowledge the unusual observation that the selfed *ire1a ire1b ire1c/+* FLAG IRE1B and *ire1a ire1b ire1c/+*FLAG IRE1B Δ LD do not produce *ire1c* homozygote progeny and offer a reasonable alternative explanation with their results in Figure S3 and in the discussion when compared to the Δ LD IRE1B CRISPR/Cas9 mutants #2-5 and #9-6.

It should be noted that the authors observations that no visible pIRE1C:GUS staining was observed in vegetative tissues runs contradictory to the observed growth defect phenotype in the *ire1a ire1b ire1c/+* plants which do not look to be at the reproductive stage. After flowering, IRE1C expression in reproductive tissues could hypothetically lead to systemic signals which could affect vegetative growth, but could the authors offer an alternative explanation in the discussion as to how a plant without any tissues which express IRE1C, display such dramatic effects from the genetic ablation of one IRE1C allele? Is it conceivable that genetic compensation by increased expression of IRE1C in the *ire1a ire1b* double mutant could be reducing developmental growth defects in the *ire1a ire1b* mutant? To resolve these contradictions and simultaneously demonstrate the transcript knockout status which was demonstrated previously for *ire1a* and *ire1b* in Nagashima et al 2011. but yet to be shown for *ire1c* (Salk_204405) in this manuscript, an expression analysis of IRE1C in WT, *ire1a*

ire1b, *ire1c*, and *ire1a ire1b ire1c/+* via qRT-PCR in leaves, shoot meristem, roots, using whole floral tissue (like that in Figure S3A) as a reference point would help. Although not expressly required, this experiment (accomplished in ~ 1 month) could strengthen the overall manuscript without requiring the generation of additional transgenic GUS lines.

According to your suggestion, we performed qRT-PCR of *IRE1C* in young seedlings and flower buds, which are shown in Figs. S3E and S3F, respectively. The seedlings grown in MS plate containing 1% sucrose, on which we can distinguish between *ire1a ire1b +/+* and *ire1a ire1b ire1c/+* (additional Fig. S3D), are vegetative stage. In the seedling tissues, *IRE1C* mRNA was expressed in *ire1a ire1b ire1c/+* at a lower level as compared to *ire1a ire1b +/+*, even though there were no significant differences ($p > 0.05$; Fig. S3E). In the flower bud tissues, *IRE1C* mRNA expression in *ire1a ire1b ire1c/+* was significantly lower than that in *ire1a ire1b +/+* (Fig. S3F). In the qRT-PCR analysis, Ct values of *IRE1C* were high (29-32) in seedlings, suggesting the *IRE1C* mRNA abundance in vegetative tissues is quite low. This is consistent with the public microarray data, and the fractional *IRE1C* expression probably makes difficult to detect visible GUS staining of p*IRE1C*:GUS in vegetative tissues. We speculate that the low *IRE1C* expression in vegetative tissues causes incomplete dominance (or semi-dominant) phenotype (PMID: 8038607). Additionally, unexpected *IRE1C* mRNA expressions were observed in WT plants (Figs. S3E, S3F). This may be due to the fact that *IRE1A* and *IRE1B* deficiencies extensively alter gene expression profiles (Nagashima *et al* 2011).

4. Different IRE1 Activation states by saturated fatty acids in the presence or absence of the IRE1B sensor domain (Partially supported).

On the whole the conclusion reached by the authors that glycerol treatment induces increases in saturated fatty acid content and concomitant IRE1 activation leading to increases the levels of spliced bZIP60 dependent upon the both the luminal and cytoplasmic functions of IRE1A and IRE1B is fully supported by Figures 6A, 6B, 6C, and most strongly supported by 6SA and 6SB.

However, the authors' assertion that RIDD activity is restored by IRE1B Δ LD, a conclusion central to the overall narrative of the manuscript, could be more strongly supported by adopting a more consistent experimental protocol and presenting more experimental replicates. Discrepancies between results in the semi-quantitative data found in figures 6E, 6F (which do not present any experimental replicates) and the quantitative data from Figure S4D, strongly subtract from reader confidence. Cordycepin, a transcription inhibitor is used to demonstrate that degradation of RIDD targets by IRE1A and IRE1B is an important determinant of the overall transcript level. While the clear decrease of PRX34 transcript levels is observed in figure 6E between 0 and 2 hours, a corresponding decrease in 6F is not readily discernable. PR4 degradation over 0-5 hours of cordycepin treatment is not easily recognized. Importantly, the PR4 and PRX34 0h and 2 hour bands

from the FLAG IRE1 Δ LD samples in Figure 6F look much closer in strength to PR4 and PRX34 0h and 2 hour bands from the ire1a ire1b samples than FLAG-IRE1B as suggested in the text. A qPCR experiment quantifying transcript levels of PR4 in WT, ire1a ire1b, FLAG-IRE1B WT, ire1a ire1b, FLAG-IRE1B K821A ire1a ire1b and FLAG- IRE1 Δ LD ire1a ire1b plants treated with glycerol for 3 days and subsequently treated with cordycepin for 0, 2, and 5 would be sufficient to replace both figures 6E and 6F and strongly support the authors' conclusion.

According to your suggestion, we performed qRT-PCR to detect RIDD activity under glycerol treatment. In the case of the detection for mRNA degradation under Tm/DTT treatments, pre-treatment with cordycepin was done 1 hour before the Tm/DTT treatments. However, glycerol treatment cannot use this procedure because the treatment needs 3 days for IRE1 activation. We therefore compared glycerol-treated samples between 3 hours and 1 hour of cordycepin treatments to detect mRNA degradation. The results of the qRT-PCR were shown in Fig. 6F, instead of the RNA gel blot panel. Consequently, the RIDD activity was lower than that induced by DTT and Tm (Mishiba et al., PNAS 2013), and we discussed in the text (P10, line 21-23).

Additional Comments:

Introduction

Paragraph 1, Line 7: Suggested rewording: Under ER stress IRE1 senses ER luminal unfolded proteins, ultimately leading to IRE1 dimerization, autophosphorylation and RNase activation.

Paragraph 2, Line 6: Suggested rewording: The disparate phenotypic consequences of IRE1 mutation between Arabidopsis and rice...

Paragraph 3, Line 6: Suggested rewording: ...presumed that unfolded protein-independent mechanisms allow cells to preemptively adapt their.....

Discussion

Paragraph 2, Line 4: Suggested rewording:due to low expression of the IRE1C transcript. IRE1C expression is strongest in the anther.....

Thank you for your suggestion. We rewrote all the sentences (P2, line 34-36; P3, line 13-15; P3, line 21-24; P8, line 34-37).

September 27, 2019

RE: Life Science Alliance Manuscript #LSA-2019-00459-TR

Dr. Kei-ichiro Mishiba
Osaka Prefecture University
Graduate School of Life and Environmental Sciences
Gakuen 1-1
Nakaku
Sakai, Osaka 599-8531
Japan

Dear Dr. Mishiba,

Thank you for submitting your revised manuscript entitled "Unfolded protein-independent IRE1 activation contributes to developmental processes in Arabidopsis". As you will see, the reviewers appreciate the introduced changes and we would thus be happy to publish your paper in Life Science Alliance pending final minor revisions:

- please address the remaining comment of reviewer #1
- I would like to encourage you to also add dividing lines between the blots in Figures 3B, 3C, 6E, 6D, 6G

A. FINAL FILES:

-- Summary blurb (enter in submission system): A short text summarizing in a single sentence the study (max. 200 characters including spaces). This text is used in conjunction with the titles of

papers, hence should be informative and complementary to the title. It should describe the context and significance of the findings for a general readership; it should be written in the present tense and refer to the work in the third person. Author names should not be mentioned.

B. MANUSCRIPT ORGANIZATION AND FORMATTING:

Sincerely,

Reviewer #1 (Comments to the Authors (Required)):

The authors have done a good job in addressing the concerns raised by the reviewers.

Even though I think that the new data on the mRNA level are less informative than immunoblots, which would report on the more relevant protein level, my main concerns were addressed by the new Figure 3B.

I think it would be important for the reader to indicate either in the main text or in the M&M section which types of lipids (TAGs, phospholipids, etc.) contribute to the fatty acid analysis used in this study. This should be stated in just one extra sentence.

That's it. Very good work!

Reviewer #2 (Comments to the Authors (Required)):

In their modifications to the manuscript "Unfolded protein-independent IRE1 activation contributes to multifaceted developmental processes in Arabidopsis", Mishiba et al. have thoroughly addressed the previous points raised by the reviewers. The revised version is of publishable quality and represents a significant advance in the field.

September 30, 2019

RE: Life Science Alliance Manuscript #LSA-2019-00459-TRR

Dr. Kei-ichiro Mishiba
Osaka Prefecture University
Graduate School of Life and Environmental Sciences
Gakuen 1-1
Nakaku
Sakai, Osaka 599-8531
Japan

Dear Dr. Mishiba,

Thank you for submitting your Research Article entitled "Unfolded protein-independent IRE1 activation contributes to developmental processes in Arabidopsis". It is a pleasure to let you know that your manuscript is now accepted for publication in Life Science Alliance. Congratulations on this interesting work.

*****IMPORTANT:** If you will be unreachable at any time, please provide us with the email address of an alternate author. Failure to respond to routine queries may lead to unavoidable delays in publication.*******

DISTRIBUTION OF MATERIALS:

Again, congratulations on a very nice paper. I hope you found the review process to be constructive and are pleased with how the manuscript was handled editorially. We look forward to future exciting

submissions from your lab.

Sincerely,
